# Heterogeneous contributions of change in population distribution of body mass index to change in obesity and underweight

**NCD Risk Factor Collaboration (NCD-RisC)\***

**Abstract** From 1985 to 2016, the prevalence of underweight decreased, and that of obesity and severe obesity increased, in most regions, with significant variation in the magnitude of these changes across regions. We investigated how much change in mean body mass index (BMI) explains changes in the prevalence of underweight, obesity, and severe obesity in different regions using data from 2896 population-based studies with 187 million participants. Changes in the prevalence of underweight and total obesity, and to a lesser extent severe obesity, are largely driven by shifts in the distribution of BMI, with smaller contributions from changes in the shape of the distribution. In East and Southeast Asia and sub-Saharan Africa, the underweight tail of the BMI distribution was left behind as the distribution shifted. There is a need for policies that address all forms of malnutrition by making healthy foods accessible and affordable, while restricting unhealthy foods through fiscal and regulatory restrictions.

**\*For correspondence:**
majid.ezzati@imperial.ac.uk;
s.filippi@imperial.ac.uk

**Group author details:**
NCD Risk Factor Collaboration
(NCD-RisC) See page 11

**Competing interests:** The
authors declare that no
competing interests exist.

**Reviewing editor:** Christine M
Friedenreich, University of
Calgary, Canada

## Introduction

Underweight as well as obesity can lead to adverse health outcomes (*Prospective Studies Collaboration et al., 2009*; *Global BMI Mortality Collaboration, 2016*; *Emerging Risk Factors Collaboration et al., 2011*). For at least four decades, the prevalence of underweight has decreased, and that of obesity has increased, in most countries with significant variation in the magnitude of these changes across regions of the world (*NCD Risk Factor Collaboration (NCD-RisC), 2017a*; *NCD Risk Factor Collaboration (NCD-RisC), 2019*).

A shift in the whole distribution of body mass index (BMI) would simultaneously affect mean BMI as well as the prevalence of underweight and obesity (*Razak et al., 2018*; *Rose and Day, 1990*). In contrast, changes in the shape of BMI distribution – for example, widening or narrowing of the BMI distribution, becoming more or less skewed, or having a thinner or thicker tail – would affect the prevalence of underweight and obesity with only small impacts on the population mean, as shown schematically in *Figure 1*. Understanding these two mechanisms is essential as they may require different public health and clinical responses (*Penman and Johnson, 2006*). But it is unclear how much the two mechanisms have contributed to the observed decline in underweight and rise in obesity in different world regions.

Some studies have investigated whether the rise in obesity or the decrease of underweight over time, or differences across countries, were due to a shift in BMI distribution versus changes in the low- or high-BMI tails of the distribution (*Razak et al., 2018*; *Wang et al., 2007*; *Wagner et al., 2019*; *Vaezghasemi et al., 2016*; *Razak et al., 2013*; *Popkin and Slining, 2013*; *Popkin, 2010*; *Peeters et al., 2015*; *Ouyang et al., 2015*; *Monteiro et al., 2002*; *Midthjell et al., 2013*; *Lebel et al., 2018*; *Khang and Yun, 2010*; *Helmchen and Henderson, 2004*; *Hayes et al., 2015*; *Green et al., 2016*; *Flegal and Troiano, 2000*; *Stenholm et al., 2015*; *Hayes et al., 2017*; *Flegal et al., 2012*; *Bovet et al., 2008*). Most of these studies focused on a single or small number of countries over relatively short durations or covered only one sex, a narrow age group, or specific

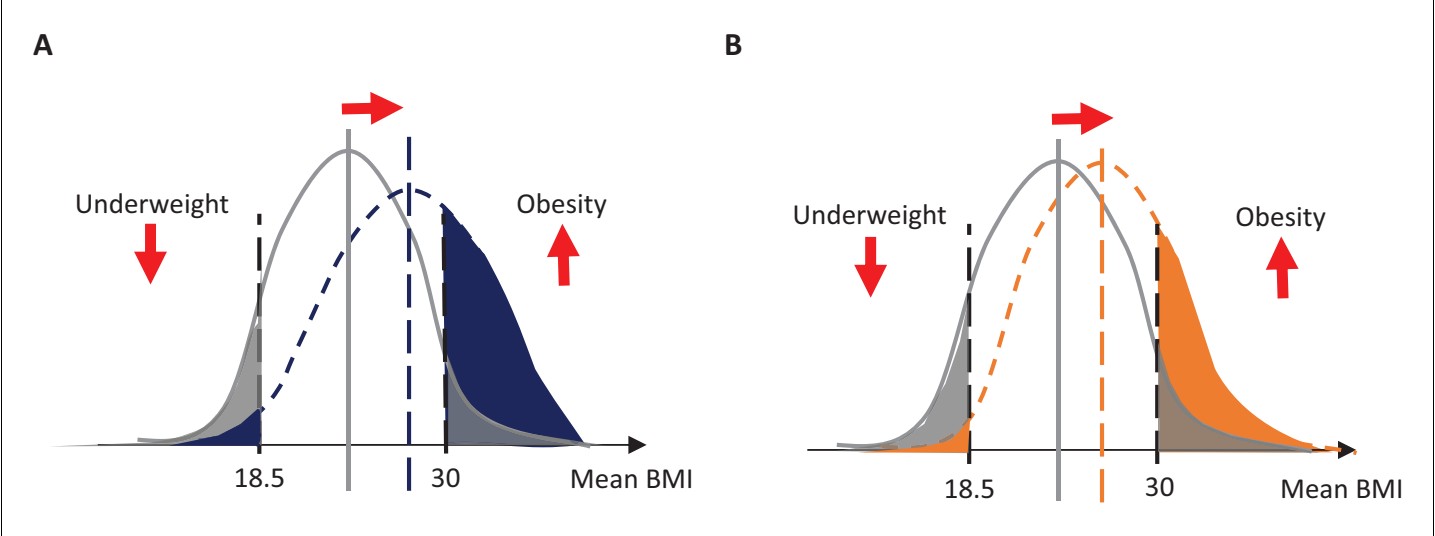

**Figure 1.** Schematic diagram of contribution of change in mean body mass index (BMI) to change in total prevalence of underweight or obesity. (**A**) Change in the prevalence of underweight and obesity if the distribution shifts, represented by a change in its mean and its shape. In this example, the change (shown as the difference between blue and gray) results in a small decrease of underweight and a large increase in obesity. (**B**) Change in the prevalence of underweight and obesity when only mean BMI changes (shown as the difference between orange and gray), without a change in the shape of the distribution.

social groups. To understand whether weight gain occurs across all BMI levels or disproportionately affects the underweight or obese segments of the distribution, and how this phenomenon varies geographically, there is a need for a population-based study that simultaneously investigates both underweight and obesity in relation to mean BMI in different regions of the world. We used a comprehensive global database to investigate how much change in mean BMI can explain the corresponding changes in prevalence of adults with underweight (defined as BMI <18.5 kg/m$^2$), total obesity (BMI ≥30 kg/m$^2$), and severe obesity (BMI ≥35 kg/m$^2$) over three decades from 1985 to 2016 in different regions of the world.

## Results

### Data sources

The Non-Communicable Disease Risk Factor Collaboration (NCD-RisC) database contains 2896 population-based studies conducted from 1985 to 2019 with height and weight measurements of 187 million participants. Of these, 2033 studies had measurements of height and weight on 132.6 million participants aged 20–79 years. The number of studies with participants aged 20–79 years in different regions ranged from 53 in Oceania to 637 in the high-income western region. The number of data sources by country is shown in *Figure 2*. The list of data sources and their characteristics is provided in *Supplementary file 4*.

### Change in mean BMI and prevalence of underweight, obesity, and severe obesity by region

In 2016, the age-standardised prevalence of underweight was highest (>16% in different sex-age groups) in South Asia; it was <2.5% in Central and Eastern Europe; the high-income western region; Latin America and the Caribbean; Oceania; and Central Asia, the Middle East, and North Africa for most age and sex groups. The age-standardised prevalence of obesity was highest (>24%) in these same regions for most age and sex groups. It was lowest (<7%) in men and women from South Asia; the high-income Asia Pacific region; and men from sub-Saharan Africa. The age-standardised prevalence of severe obesity was highest (12–18%) in women aged 50–79 years from Central Asia, the Middle East, and North Africa; the high-income western region; Central and Eastern Europe; and

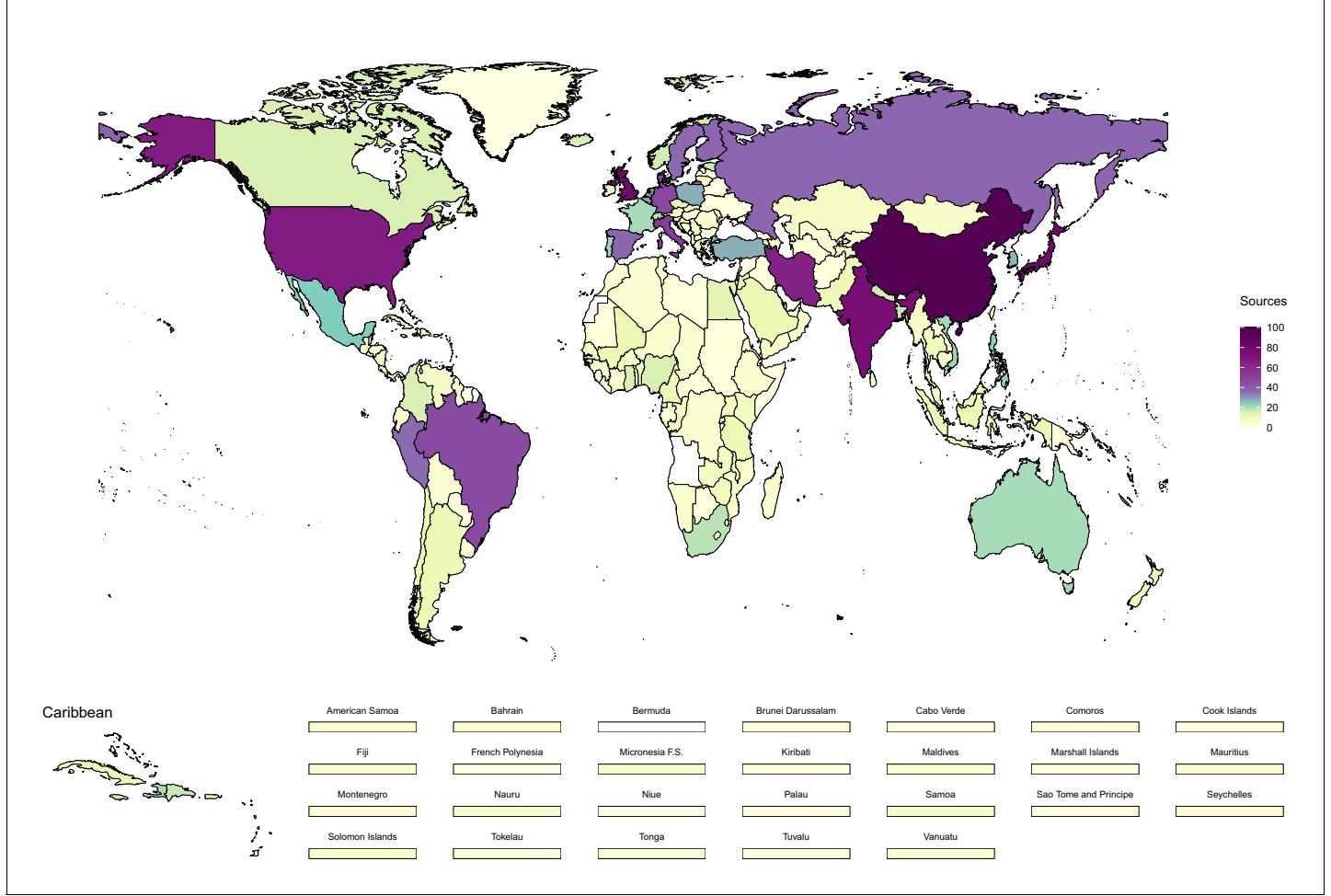

**Figure 2.** Number of data sources with participants aged 20-79 years.

Latin America and the Caribbean. It was lowest (<2%) in South Asia; East and Southeast Asia; the high-income Asia Pacific region; and men in sub-Saharan Africa.

From 1985 to 2016, age-standardised mean BMI increased by 1–4 kg/m$^2$ in all regions, with the exception of women in the high-income Asia Pacific region and Central and Eastern Europe whose mean BMI changed by less than 1 kg/m$^2$ (*Figure 3*). The prevalence of underweight decreased or stayed unchanged and that of obesity and severe obesity increased from 1985 to 2016 in all regions, with the exception of an increase in the prevalence of underweight in younger women in the high-income Asia Pacific region. The largest absolute decrease in underweight prevalence from 1985 to 2016 was seen in South Asia; East and Southeast Asia; and sub-Saharan Africa, where it declined by 14–35 percentage points in different age–sex groups (*Figure 4*). Nonetheless, underweight prevalence remained higher in these three regions than elsewhere in 2016. Prevalence of underweight changed only marginally in regions such as Central and Eastern Europe and the high-income western region, where prevalence was already low in 1985.

The largest absolute increase in obesity prevalence from 1985 to 2016 occurred in Central Asia, the Middle East, and North Africa; the high-income western region; Latin America and the Caribbean; Oceania (women); and Central and Eastern Europe (men) (*Figure 4*). Women in these regions also experienced the largest increase in severe obesity prevalence, along with men in the high-income western region. In these regions and sexes, obesity prevalence increased by 16–24 percentage points in different age groups, and severe obesity increased by 5–13 percentage points. The increase in obesity was less than five percentage points in the high-income Asia Pacific region; South

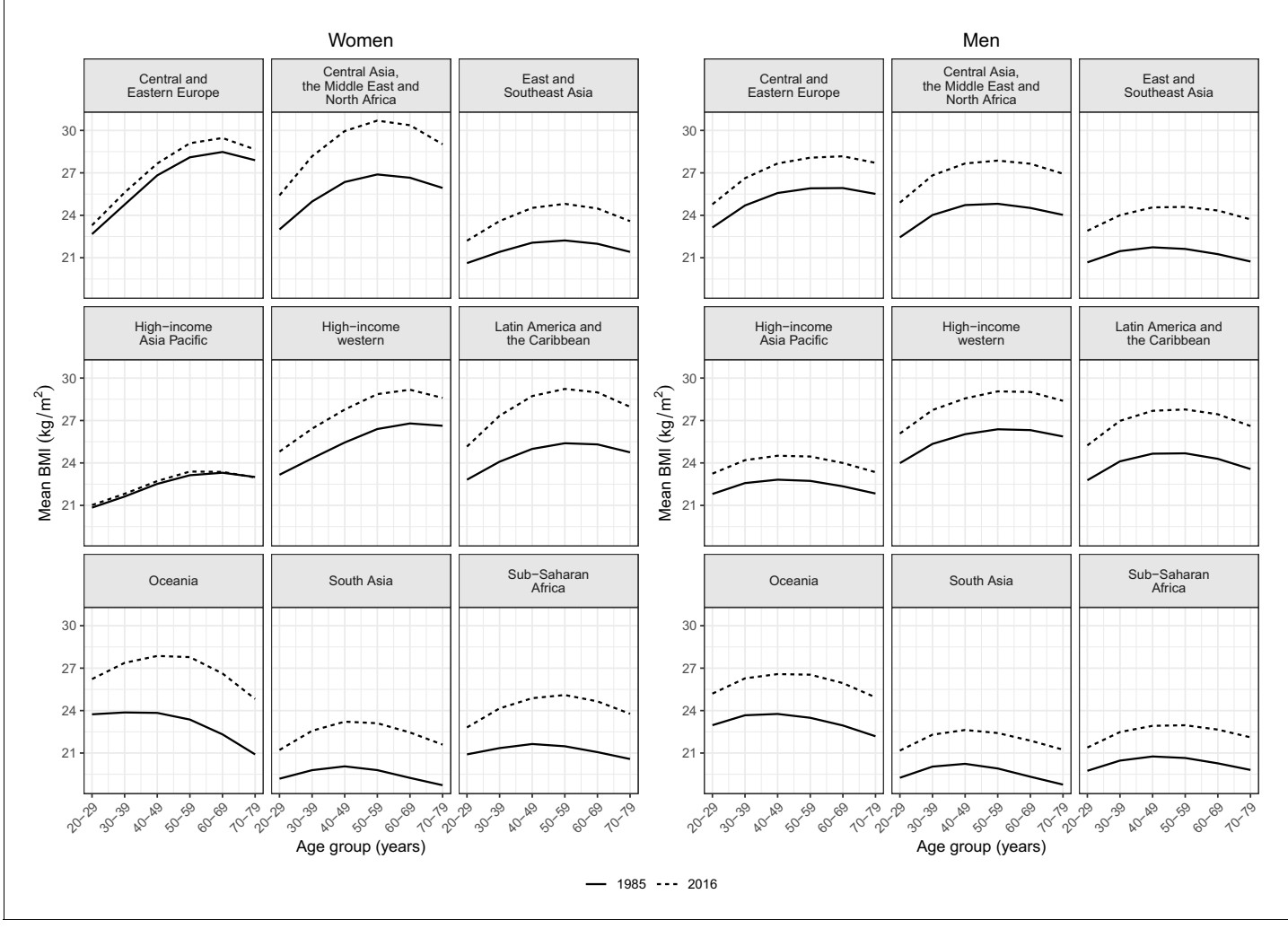

**Figure 3.** Change in mean body mass index (BMI) from 1985 to 2016 by region, sex, and age group.

Asia; and in men in sub-Saharan Africa; in the same regions, along with East and Southeast Asia, the increase in severe obesity was less than two percentage points.

## Associations of underweight, obesity, and severe obesity prevalence with mean BMI

There was a strong association between the prevalence of underweight, obesity, and severe obesity with mean BMI as measured by R-squared of the regressions of prevalence on mean (*Supplementary files 1* and *2*). These indicate that 93% (men) and 96% (women) of variation in obesity, and between 83% and 92% of variation in underweight and severe obesity, were explained by mean BMI and other variables (year, region, and age group) in the regression models. The coefficients of the mean BMI terms represent the changes in (probit-transformed) prevalence associated with a unit change in mean BMI, and their interactions with region represent variations in this association across regions. For all three outcomes, the association of prevalence with mean BMI varied across regions.

The inter-regional variation in the prevalence–mean association was stronger for obesity and severe obesity than underweight, as seen in larger inter-regional range of the interaction terms. The extent to which prevalence changes with any variation in mean BMI in each region is an outcome of the main BMI term and its interaction with region; to be epidemiologically interpretable, this will have to be converted from probit-transformed to original prevalence scale. For example, in the year

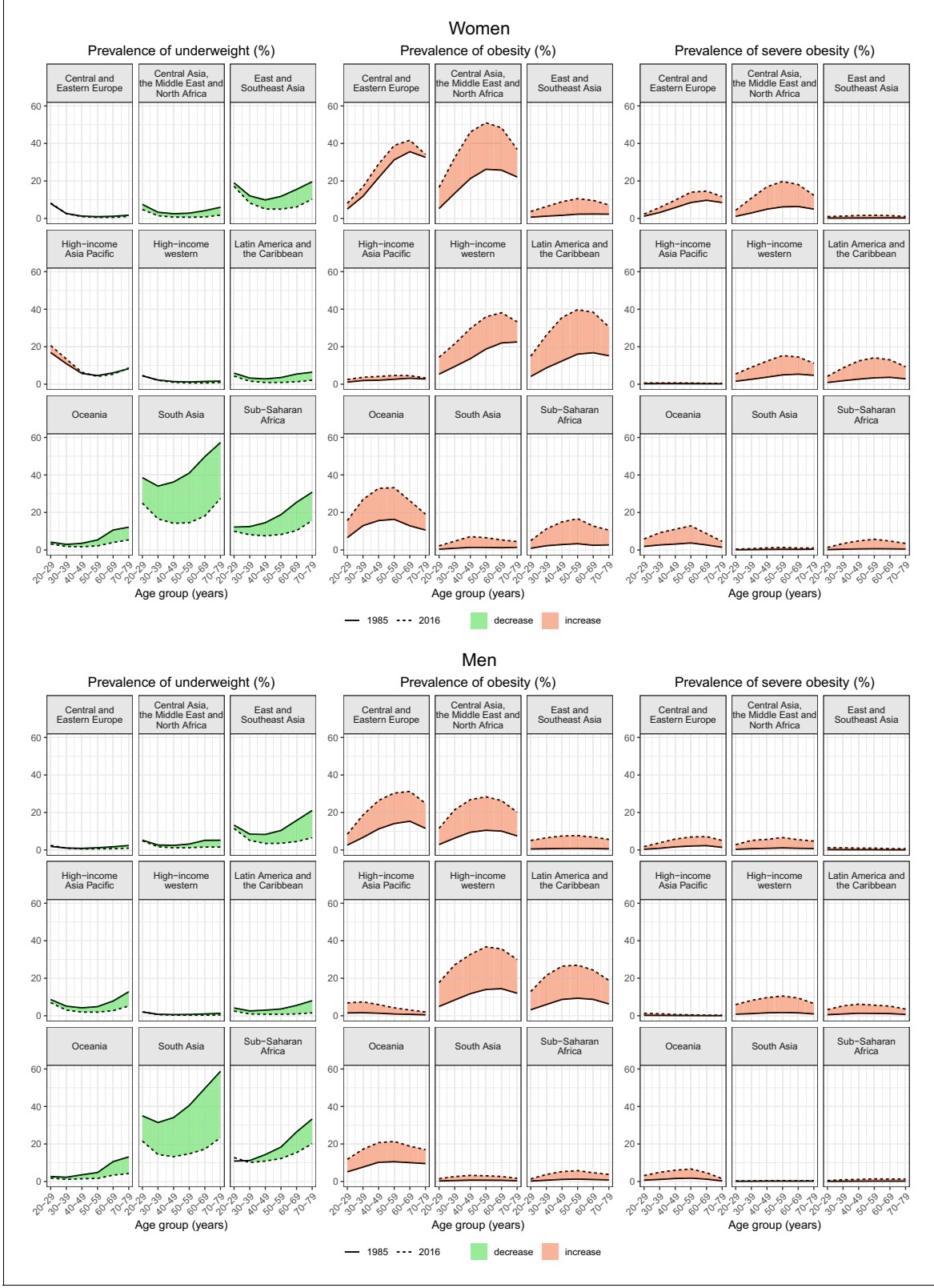

**Figure 4.** Change in prevalence of underweight, obesity, and severe obesity from 1985 to 2016 by region, sex, and age group.

2016, for women aged 50–59 years, at a mean BMI of 25 kg/m$^2$ (which was approximately the global age-standardised mean level of BMI) (*NCD Risk Factor Collaboration (NCD-RisC), 2017a*), prevalence of underweight would have varied by seven percentage points across regions, being lowest in Central and Eastern Europe and highest in sub-Saharan Africa; a unit increase in mean BMI would have been associated with a relative change in prevalence ranging from −49% in the high-income Asia Pacific region to −14% in Oceania. Also for women aged 50–59 years and a mean BMI of 25 kg/m$^2$, the prevalence of obesity and severe obesity would both have been the highest in Oceania and the lowest in the high-income Asia Pacific region, with a difference of 12 and 6 percentage points, respectively, for the two outcomes; a unit increase in mean BMI would have been associated with a relative change ranging from 21% to 46% for obesity and from 30% to 59% for severe obesity, the smallest change for both being in Oceania and the largest in East and Southeast Asia. There was similar inter-regional variation in the other year–age–sex strata.

### Contribution of mean BMI to changes in underweight and obesity prevalence

The rise in mean BMI accounted for >82% of the decline in underweight in different age–sex groups in South Asia, where underweight prevalence declined by over 16 percentage points for all age–sex groups (*Figure 5*). The remainder of the decline was due to change in the shape of the BMI distribution which reduced underweight prevalence beyond the effects of the population mean. In contrast, in sub-Saharan Africa and East and Southeast Asia, the total change in prevalence of underweight (3–12 percentage points) was 20–80% less than what was expected based on the increase in mean BMI (*Figure 5*). In other words, in these regions the underweight tail of the BMI distribution was left behind as the distribution shifted.

Where obesity increased the most – Central Asia, the Middle East, and North Africa; Latin America and the Caribbean; and the high-income western region – the rise in mean BMI accounted for over three quarters of the increase in different age–sex groups (*Figure 5*). In Oceania, the actual rise in prevalence of obesity (8–14 percentage points for all age–sex groups) was about two-thirds to one-half of what would have been expected by the observed increase in mean BMI in men and women (*Figure 5*). Change in mean BMI consistently accounted for a smaller share of the change in severe obesity than it did for change in total obesity. Specifically, in regions where prevalence of severe obesity changed by more than one percentage point, the contribution of change in mean BMI to change in severe obesity in different regions was 53–90% of the corresponding contribution for total obesity (*Figure 5*).

In other regions, the change in the prevalence of underweight, obesity, or severe obesity was too small for the contribution of change in mean BMI to be epidemiologically relevant (*Figure 5*).

## Discussion

We found that the trends in the prevalence of underweight, total obesity, and, to a lesser extent, severe obesity are largely driven by shifts in the distribution of BMI, with smaller contributions from changes in the shape of the distribution. The notable exceptions to this pattern were the decline in the prevalence of underweight in East and Southeast Asia and sub-Saharan Africa and the rise of obesity in Oceania, which were both smaller than expected based on change in mean BMI.

Our results are consistent with a recent cross-sectional study (*Razak et al., 2018*) using data from women in low- and middle-income countries that found a strong association between mean BMI and prevalence of obesity, and a moderate association between mean BMI and prevalence of underweight. Being cross-sectional, this study did not consider changes over time, as we have. Our results are also consistent with another study which found that changes in median BMI contributed more than 75% to the increase in obesity in the USA from 1980 to 2000 (*Helmchen and Henderson, 2004*).

Previous studies used one or more approaches to investigate changes in population BMI distribution: some analysed percentiles of the BMI distribution (*Wagner et al., 2019*; *Vaezghasemi et al., 2016*; *Razak et al., 2013*; *Popkin and Slining, 2013*; *Popkin, 2010*; *Peeters et al., 2015*; *Ouyang et al., 2015*; *Lebel et al., 2018*; *Hayes et al., 2015*), others focused on the change in prevalence above or below pre-specified BMI thresholds (*Wang et al., 2007*; *Razak et al., 2013*; *Popkin, 2010*; *Peeters et al., 2015*; *Ouyang et al., 2015*; *Khang and Yun, 2010*; *Flegal and Troiano,*

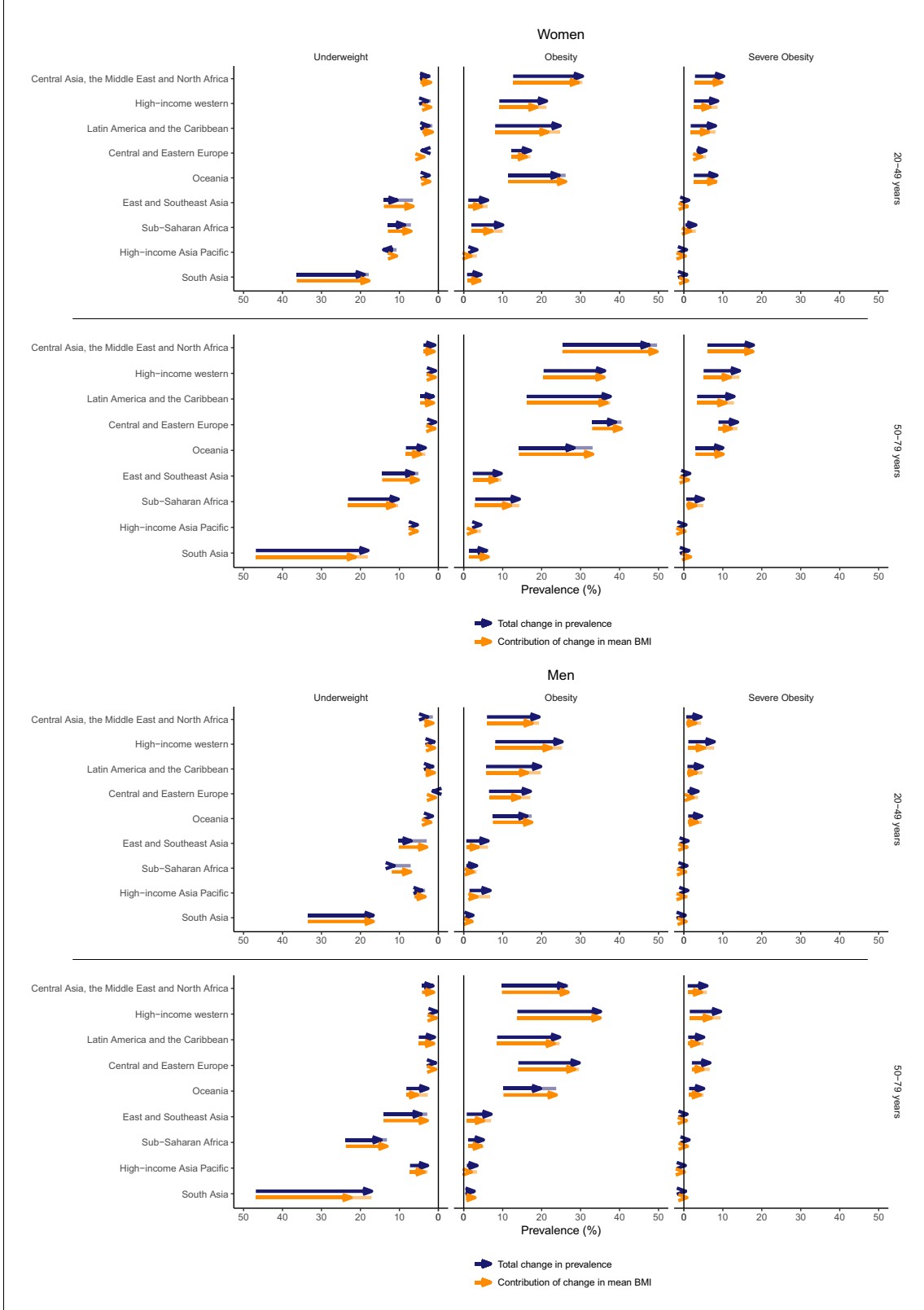

**Figure 5.** Contribution of change in mean body mass index (BMI) to total change from 1985 to 2016 in prevalence of underweight, obesity, or severe obesity by region, sex, and age group. Blue arrows show the total change in prevalence of underweight, obesity, or severe obesity. Orange arrows show the contribution of change in mean BMI to the change in prevalence. The difference between these two arrows is shown with a line, whose colour follows the shorter arrow.

*2000*), or evaluated how the shape of the BMI distribution has changed via examining metrics such as standard deviation and skewness (*Peeters et al., 2015*; *Ouyang et al., 2015*; *Lebel et al., 2018*; *Khang and Yun, 2010*; *Hayes et al., 2015*; *Flegal and Troiano, 2000*). Most of these studies reached the same conclusion as our study that, as the BMI distribution shifts upwards, the prevalence of underweight declines somewhat more slowly than the prevalence of obesity rises.

Our study has strengths in scope, data, and methods: the strengths of our study include presenting the first global analysis of how much the rise in mean BMI versus changes in the shape of its distribution influenced changes in both underweight and obesity prevalence. We used an unprecedented amount of data from different regions covering three decades and used only measured data on height and weight to avoid biases in self-reported data.

As with all global analyses, our study has some limitations. Despite using the most comprehensive global collection of population-based studies to date, some regions, especially Oceania and sub-Saharan Africa, had less data, especially early in our analysis period. Further, given the large number of age, sex, and region subgroups of population in our analysis, and its long duration, it was not possible to visually explore how the shape of BMI distribution has changed in the underweight and obesity ranges where changes in the mean did not fully explain change in prevalence. Finally, there are variations in characteristics such as response rate and measurement protocol across studies. Some of these, such as exclusion of studies with self-reported height and weight, were a part of our inclusion and exclusion criteria. Others may affect population mean or prevalence.

The finding that the majority of the rise in the prevalence of obesity from 1985 to 2016 is mostly the result of a distributional shift points towards an important role for societal drivers, including lower availability and higher price of healthy and fresh foods compared to caloric-dense and nutrient-deficient foods (*Swinburn et al., 2011*), and mechanisation of work and motorisation of transport throughout the world that have reduced energy expenditure in populations around the world (*NCD Risk Factor Collaboration (NCD-RisC), 2019*; *Ng and Popkin, 2012*). First, although there is a genetic component to BMI at the individual level (*Silventoinen et al., 2017*; *Silventoinen et al., 2016*; *Locke et al., 2015*; *Brandkvist et al., 2019*), genetics explain only a small part of changes over time, especially when people have access to healthy food and living environment. When the environment becomes more obesogenic, some people or population subgroups may gain more weight than others, implying that the environment remains the main contributor (*Brandkvist et al., 2019*). This interplay of genetic predisposition and changes in the environment might account for some of the excess rise in obesity and severe obesity beyond the effect of distributional shift alone (*Brandkvist et al., 2019*). The exception observed in Oceania is possibly because in 1985 obesity prevalence in this region was already so high (*NCD Risk Factor Collaboration (NCD-RisC), 2017a*) that the rise in BMI did not change overall obesity status (but there was a substantial increase in those with severe obesity, mostly accounted for by the change in mean BMI). The smaller decline in underweight than expected in sub-Saharan Africa and East and Southeast Asia may be because underweight is associated with lower socioeconomic status, food insecurity, and for sub-Saharan Africa widening difference between rural and urban BMI levels which is different from other regions (*NCD Risk Factor Collaboration (NCD-RisC), 2019*; *Brandkvist et al., 2019*; *Di Cesare et al., 2015*; *Subramanian and Smith, 2006*; *Di Cesare et al., 2013*). If the benefits of economic development do not sufficiently reach the poor, they remain nutritionally vulnerable, as has been seen for height and weight during childhood and adolescence (*NCD Risk Factor Collaboration (NCD-RisC), 2020*; *Subramanyam et al., 2011*; *Sanchez and Swaminathan, 2005*; *Pongou et al., 2006*; *Haddad, 2003*; *Stevens et al., 2012*). Together with the rise in mean BMI and obesity (and short stature which is not a topic of this paper but addressed in other studies) (*NCD Risk Factor Collaboration (NCD-RisC), 2020*; *Stevens et al., 2012*; *NCD Risk Factor Collaboration (NCD-RisC), 2016a*), this creates a double burden of malnutrition (*Popkin et al., 2020*).

In summary, we found that the worldwide rise in obesity and the decline in underweight are primarily driven by the shift in the population distribution of BMI. At the same time, there is an evidence of both excess obesity, and especially severe obesity, and persistent underweight beyond the distributional shift in some regions, which may be related to growing social inequalities that restrict access to healthy foods in those at highest risk of undernutrition (*Popkin et al., 2020*; *Wells et al., 2020*; *Darmon and Drewnowski, 2015*). The response to these trends must motivate 'double-duty actions' that prevent and tackle all forms of malnutrition through both fiscal and regulatory restrictions on unhealthy foods, and making healthy foods available, accessible, and affordable especially

to those at high risks of underweight and obesity (*Powell et al., 2013*; *Hawkes et al., 2020*; *Bleich et al., 2017*).

# Materials and methods

## Study design

Our aim was to quantify, for all regions of the world, how much of the change in prevalence of underweight (defined as BMI <18.5 kg/m$^2$), (total) obesity (BMI $\geq$30 kg/m$^2$), and severe obesity (BMI $\geq$35 kg/m$^2$) among men and women aged 20–79 years from 1985 to 2016 could be accounted for by change in mean BMI. In the first step, we used data from a global database of human anthropometry to estimate the associations of the prevalence of underweight, prevalence of obesity, or prevalence of severe obesity with population mean BMI, including how the association varies in relation to age group and region. We then used the fitted association to estimate the contribution of change in the population mean BMI to change in the prevalence of underweight, obesity, or severe obesity in different regions.

## Data sources

In the first step of the analysis, we estimated the prevalence-mean associations, using data from a comprehensive database on cardiometabolic risk factors collated by NCD-RisC as described below. In the second step, we needed consistent estimates of mean BMI for all regions. For this purpose, we used data from a recent comprehensive analysis of worldwide trends in mean BMI from 1985 to 2016 (*NCD Risk Factor Collaboration (NCD-RisC), 2017a*) which had fitted a Bayesian hierarchical model to the NCD-RisC data.

Data in the NCD-RisC database were obtained from publicly available multi-country and national measurement surveys (e.g., Demographic and Health Surveys, WHO-STEPwise approach to Surveillance [STEPS] surveys, and those identified via the Inter-University Consortium for Political and Social Research and European Health Interview and Health Examination Surveys Database). With the help of the World Health Organization (WHO) and its regional and country offices as well as the World Heart Federation, we identified and accessed population-based survey data from national health and statistical agencies. We searched and reviewed published studies as detailed previously (*NCD Risk Factor Collaboration (NCD-RisC), 2017a*) and invited eligible studies to join NCD-RisC, as we did with data holders from earlier pooled analysis of cardiometabolic risk factors (*Finucane et al., 2011*; *Farzadfar et al., 2011*; *Danaei et al., 2011a*; *Danaei et al., 2011b*).

## Data inclusion and exclusion

We carefully checked that each data source met our inclusion criteria as listed below:

- measurement data on height and weight were available;
- study participants were 5 years of age and older (as described earlier data used here were for those 20–79 years);
- data were collected using a probabilistic sampling method with a defined sampling frame;
- data were from population samples at the national, sub-national (i.e., covering one or more sub-national regions, with more than three urban or five rural communities), or community level; and
- data were from the countries and territories listed in *Supplementary file 3*.

We excluded all data sources that only used self-reported weight and height without a measurement component because these data are subject to biases that vary with geography, time, age, sex, and socioeconomic characteristics (*Tolonen et al., 2014*; *Hayes et al., 2011*; *Ezzati et al., 2006*). We also excluded data on population subgroups whose anthropometric status may differ systematically from the general population, including

- studies that had included or excluded people based on their health status or cardiovascular risk;
- studies whose participants were only ethnic minorities;
- specific educational, occupational, or socioeconomic subgroups, with the exception noted below; and
- those recruited through health facilities, with the exception noted below.

We included school-based data in countries and age–sex groups with enrolment of 70% or higher. We also included data whose sampling frame was health insurance schemes in countries where at least 80% of the population were insured. Finally, we included data collected through general practice and primary care systems in high-income and Central European countries with universal insurance because contact with the primary care systems tends to be as good as or better than the response rates for population-based surveys. The list of data sources with participants aged 20–79 years and their characteristics is provided in *Supplementary file 4*, with additional information in *Source data 1*.

Duplicate data were identified by comparing studies from the same country and year, and then discarded. All NCD-RisC members are also periodically asked to review the list of sources from their country, to verify that the included data meet the inclusion criteria and are not duplicates, and to suggest additional sources. The NCD-RisC database is continuously updated through all the above routes. For each data source, we recorded the study population, sampling approach, years of measurement, and measurement methods. Only population-representative data were included, and these were assessed in terms of whether they covered the whole country, multiple sub-national regions, or one or a small number of communities, and whether rural, urban, or both participants were included. All submitted data were checked independently by at least two persons. Questions and clarifications were discussed with NCD-RisC members and resolved before data were incorporated in the database.

We calculated mean BMI and the associated standard errors by sex and age. All analyses incorporated sample weights and complex survey design, when applicable, in calculating summary statistics, with computer code provided to NCD-RisC members who requested assistance.

Additionally, summary statistics for nationally representative data from sources that were identified but not accessed via the above routes were extracted from published reports. Data were also extracted for nine STEPS surveys that were not publicly available, one Countrywide Integrated Noncommunicable Diseases Intervention survey, and five sites of the WHO Multinational MONItoring of trends and determinants in CArdiovascular disease (MONICA) project that were not deposited in the MONICA Data Centre. We also included data from a previous global data pooling study (*Finucane et al., 2011*) when they had not been accessed as described above.

Here, to estimate the association of underweight, obesity, and severe obesity prevalence with mean BMI as described below, we used data collected from 1985 to 2019 with measured height and weight among men and women aged 20–79 years, in 10-year age groups. Data that did not cover the complete 10-year age groups, for example, 25–29 or 60–64 years, were excluded. We included data from study–age–sex strata with a prevalence between 0 and 1 to allow probit transformation and with at least 25 participants in each stratum. These data were summarised into 11,652 study–age–sex-specific pairs of mean and prevalence of adults with underweight, obesity, or severe obesity.

## Statistical methods

Anonymised data from studies in the NCD-RisC database were reanalysed according to a common protocol. We calculated mean BMI and prevalence of underweight, obesity, and severe obesity by sex and age group in each study in the NCD-RisC database from 1985 to 2019. We used data through 2019 so that the prevalence–mean association is informed by as much data as possible. All calculations took into account complex survey design where relevant. We excluded study–age–sex groups with less than 25 participants because their means and prevalence have larger uncertainty.

We then estimated the relationship between probit-transformed prevalence of underweight, obesity, and severe obesity and mean BMI in a regression model, separately for each of these prevalences. The correlation coefficient between mean BMI and median BMI was $\geq 0.98$ in different age–sex groups, indicating a strong correlation between the two. In our statistical model, described below, the prevalence of underweight, obesity, or severe obesity depends on population mean BMI as well as on age group, region, and year.

All analyses were done separately for men and women. We chose a probit-transformed prevalence because it changes in an approximately linear manner as the mean changes, thus providing a better fit to the data. The regressions also included age group in 10-year bands, region and the year when the data were collected. The regions, used in previous analyses of cardiometabolic risk factors

(*NCD Risk Factor Collaboration (NCD-RisC), 2017a*; *NCD Risk Factor Collaboration (NCD-RisC), 2019*; *NCD Risk Factor Collaboration (NCD-RisC), 2020*; *NCD Risk Factor Collaboration (NCD-RisC), 2016a*; *NCD Risk Factor Collaboration (NCD-RisC), 2018*; *NCD Risk Factor Collaboration (NCD-RisC), 2017b*; *NCD Risk Factor Collaboration (NCD-RisC), 2016b*; *NCD Risk Factor Collaboration (NCD-RisC), 2016c*), were Central and Eastern Europe; Central Asia, the Middle East, and North Africa; East and Southeast Asia; high-income Asia Pacific region; high-income western region; Latin America and the Caribbean; Oceania; South Asia; and sub-Saharan Africa. Countries in each region are listed in *Supplementary file 3*. The model also included interactions between mean BMI and age group, mean BMI and region, age group and region, age group and year, and year and region. These terms allowed the prevalence–mean association to vary by age group, region, and over time. The models were fitted in statistical software R (version 4.0.2) (*Source code 1*). The coefficients of the regression models are presented in *Supplementary files 1* and *2*.

We used the fitted regressions to quantify how much of the change over time in the prevalence of underweight, obesity, or severe obesity in each region and age group can be explained by the corresponding change in mean BMI. To do so, we first used the region- and age–sex-specific mean BMI in 1985 and 2016 in the fitted association and then estimated the total change in prevalence of underweight, obesity, or severe obesity by region. The mean BMI values were from a recent comprehensive analysis of worldwide trends in mean BMI (*NCD Risk Factor Collaboration (NCD-RisC), 2017a*) and are listed in *Supplementary file 5*. We then calculated the contribution of change in mean BMI to the change in prevalence of underweight or obesity by allowing mean BMI for each age group and region to change over time, while keeping year fixed at 1985. Results were calculated by 10-year age group and then aggregated into two age bands, 20–49 and 50–79 years, by taking weighted average of age-specific results using weights from the WHO standard population (*Ahmad et al., 2001*).

## Acknowledgements

We thank WHO country and regional offices and World Heart Federation for support in data identification and access. The NCD-RisC database was funded by the Wellcome Trust. Maria LC Iurilli was supported by a Medical Research Council studentship. Sylvain Sebert received funding by the European Commission with grant agreements 633595 and 874739, respectively, for the DynaHEALTH and LongITools projects. The following contributors have deceased: Konrad Jamrozik, Altan Onat, Robespierre Ribeiro, Michael Sjöström, Agustinus Soemantri, Jutta Stieber, and Dimitrios Trichopoulos. The list of authors shows their last affiliation.

## Additional information

### Group author details

**NCD Risk Factor Collaboration (NCD-RisC)**
Maria LC Iurilli: Imperial College London, London, United Kingdom; Bin Zhou: Imperial College London, London, United Kingdom; James E Bennett: Imperial College London, London, United Kingdom; Rodrigo M Carrillo-Larco: Imperial College London, London, United Kingdom; Marisa K Sophiea: Imperial College London, London, United Kingdom; Andrea Rodriguez-Martinez: Imperial College London, London, United Kingdom; Honor Bixby: Imperial College London, London, United Kingdom; Bethlehem D Solomon: Imperial College London, London, United Kingdom; Cristina Taddei: Imperial College London, London, United Kingdom; Goodarz Danaei: Harvard TH Chan School of Public Health, Boston, United States; Mariachiara Di Cesare: Middlesex University, London, United Kingdom; Gretchen A Stevens: Independent researcher, Los Angeles, United States; Imperial College London, London, United Kingdom; Leanne M Riley: World Health Organization, Geneva, Switzerland; Stefan Savin: World Health Organization, Geneva, Switzerland; Melanie J Cowan: World Health Organization, Geneva, Switzerland; Pascal Bovet: Ministry of Health, Victoria, Seychelles; University of Lausanne, Lausanne, Switzerland; Albertino Damasceno: Eduardo Mondlane University, Maputo, Mozambique; Adela Chirita-Emandi: Victor Babes University of Medicine and Pharmacy Timisoara, Timisoara, Romania;

Alison J Hayes: University of Sydney, Sydney, Australia; Nayu Ikeda: National Institutes of Biomedical Innovation, Health and Nutrition, Tokyo, Japan; Rod T Jackson: University of Auckland, Auckland, New Zealand; Young-Ho Khang: Seoul National University, Seoul, Republic of Korea; Avula Laxmaiah: ICMR - National Institute of Nutrition, Hyderabad, India; Jing Liu: Capital Medical University Beijing An Zhen Hospital, Beijing, China; J Jaime Miranda: Universidad Peruana Cayetano Heredia, Lima, Peru; Olfa Saidi: University Tunis El Manar, Tunis, Tunisia; Sylvain Sebert: University of Oulu, Oulu, Finland; Maroje Sorić: University of Zagreb, Zagreb, Croatia; Gregor Starc: University of Ljubljana, Ljubljana, Slovenia; Edward W Gregg: Imperial College London, London, United Kingdom; Leandra Abarca-Gómez: Caja Costarricense de Seguro Social, San José, Costa Rica; Ziad A Abdeen: Al-Quds University, East Jerusalem, State of Palestine; Shynar Abdrakhmanova: National Center of Public Healthcare, Nur-Sultan, Kazakhstan; Suhaila Abdul Ghaffar: Ministry of Health, Kuala Lumpur, Malaysia; Hanan F Abdul Rahim: Qatar University, Doha, Qatar; Niveen M Abu-Rmeileh: Birzeit University, Birzeit, State of Palestine; Jamila Abubakar Garba: Usmanu Danfodiyo University Teaching Hospital, Sokoto, Nigeria; Benjamin Acosta-Cazares: Instituto Mexicano del Seguro Social, Mexico City, Mexico; Robert J Adams: Flinders University, Adelaide, Australia; Wichai Aekplakorn: Mahidol University, Nakhon Pathom, Thailand; Kaosar Afsana: BRAC James P Grant School of Public Health, Dhaka, Bangladesh; Shoaib Afzal: University of Copenhagen, Copenhagen, Denmark; Copenhagen University Hospital, Copenhagen, Denmark; Imelda A Agdeppa: Food and Nutrition Research Institute, Taguig, Philippines; Javad Aghazadeh-Attari: Urmia University of Medical Sciences, Urmia, Islamic Republic of Iran; Carlos A Aguilar-Salinas: Instituto Nacional de Ciencias Médicas y Nutrición, Mexico City, Mexico; Charles Agyemang: University of Amsterdam, Amsterdam, Netherlands; Mohamad Hasnan Ahmad: Ministry of Health, Kuala Lumpur, Malaysia; Noor Ani Ahmad: Ministry of Health, Kuala Lumpur, Malaysia; Ali Ahmadi: Shahrekord University of Medical Sciences, Shahrekord, Islamic Republic of Iran; Naser Ahmadi: Non-Communicable Diseases Research Center, Tehran, Islamic Republic of Iran; Soheir H Ahmed: University of Oslo, Oslo, Norway; Wolfgang Ahrens: University of Bremen, Bremen, Germany; Gulmira Aitmurzaeva: Republican Center for Health Promotion, Bishkek, Kyrgyzstan; Kamel Ajlouni: National Center for Diabetes, Endocrinology and Genetics, Amman, Jordan; Hazzaa M Al-Hazzaa: Princess Nourah bint Abdulrahman University, Riyadh, Saudi Arabia; Badreya Al-Lahou: Kuwait Institute for Scientific Research, Kuwait City, Kuwait; Rajaa Al-Raddadi: King Abdulaziz University, Jeddah, Saudi Arabia; Monira Alarouj: Dasman Diabetes Institute, Kuwait City, Kuwait; Fadia AlBuhairan: Aldara Hospital and Medical Center, Riyadh, Saudi Arabia; Shahla AlDhukair: King Abdullah International Medical Research Center, Riyadh, Saudi Arabia; Mohamed M Ali: World Health Organization, Geneva, Switzerland; Abdullah Alkandari: Dasman Diabetes Institute, Kuwait City, Kuwait; Ala'a Alkerwi: Luxembourg Institute of Health, Strassen, Luxembourg; Kristine Allin: Bispebjerg and Frederiksberg Hospital, Copenhagen, Denmark; Mar Alvarez-Pedrerol: Barcelona Institute for Global Health CIBERESP, Barcelona, Spain; Eman Aly: World Health Organization Regional Office for the Eastern Mediterranean, Cairo, Egypt; Deepak N Amarapurkar: Bombay Hospital and Medical Research Centre, Mumbai, India; Parisa Amiri: Research Center for Social Determinants of Health, Tehran, Islamic Republic of Iran; Norbert Amougou: UMR CNRS-MNHN 7206 Eco-anthropologie, Paris, France; Philippe Amouyel: University of Lille, France; Lille University Hospital, Lille, France; Lars Bo Andersen: Western Norway University of Applied Sciences, Sogndal, Norway; Sigmund A Anderssen: Norwegian School of Sport Sciences, Oslo, Norway; Lars Ängquist: University of Copenhagen, Copenhagen, Denmark; Ranjit Mohan Anjana: Madras Diabetes Research Foundation, Chennai, India; Alireza Ansari-Moghaddam: Zahedan University of Medical Sciences, Zahedan, Islamic Republic of Iran; Hajer Aounallah-Skhiri: National Institute of Public Health, Tunis, Tunisia; Joana Araújo: Institute of Public Health of the University of Porto, Porto, Portugal; Inger Ariansen: Norwegian Institute of Public Health, Oslo, Norway; Tahir Aris: Ministry of Health, Kuala Lumpur, Malaysia; Raphael E Arku: University of Massachusetts Amherst, Amherst, United States; Nimmathota Arlappa: ICMR - National Institute of Nutrition, Hyderabad, India; Krishna K Aryal: Abt Associates, Kathmandu, Nepal; Thor Aspelund: University of Iceland, Reykjavik, Iceland; Felix K Assah: University of Yaoundé 1, Yaoundé, Cameroon; Maria Cecília F Assunção: Federal University of Pelotas, Pelotas, Brazil; May Soe Aung: University of Medicine 1, Yangon, Myanmar; Juha Auvinen: University of Oulu, Oulu, Finland; Oulu University Hospital, Oulu, Finland; Mária Avdicová:

Banska Bystrica Regional Authority of Public Health, Banska Bystrica, Slovakia; Shina Avi: Tel-Aviv University, Tel-Aviv, Israel; Hebrew University of Jerusalem, Jerusalem, Israel; Ana Azevedo: University of Porto Medical School, Porto, Portugal; Mohsen Azimi-Nezhad: Neyshabur University of Medical Sciences, Neyshabur, Islamic Republic of Iran; Fereidoun Azizi: Research Institute for Endocrine Sciences, Tehran, Islamic Republic of Iran; Mehrdad Azmin: Non-Communicable Diseases Research Center, Tehran, Islamic Republic of Iran; Bontha V Babu: Indian Council of Medical Research, New Delhi, India; Maja Bæksgaard Jørgensen: National Institute of Public Health, Copenhagen, Denmark; Azli Baharudin: Ministry of Health, Kuala Lumpur, Malaysia; Suhad Bahijri: King Abdulaziz University, Jeddah, Saudi Arabia; Jennifer L Baker: Bispebjerg and Frederiksberg Hospital, Copenhagen, Denmark; Nagalla Balakrishna: ICMR - National Institute of Nutrition, Hyderabad, India; Mohamed Bamoshmoosh: University of Science and Technology, Sana'a, Yemen; Maciej Banach: Medical University of Lodz, Lodz, Poland; Piotr Bandosz: Medical University of Gdansk, Gdansk, Poland; José R Banegas: Universidad Autónoma de Madrid CIBERESP, Madrid, Spain; Joanna Baran: University of Rzeszów, Rzeszów, Poland; Carlo M Barbagallo: University of Palermo, Palermo, Italy; Alberto Barceló: Pan American Health Organization, Washington DC, United States; Amina Barkat: Mohammed V University de Rabat, Rabat, Morocco; Aluisio JD Barros: Federal University of Pelotas, Pelotas, Brazil; Mauro Virgílio Gomes Barros: University of Pernambuco, Recife, Brazil; Abdul Basit: Baqai Institute of Diabetology and Endocrinology, Karachi, Pakistan; Joao Luiz D Bastos: Federal University of Santa Catarina, Florianópolis, Brazil; Iqbal Bata: Dalhousie University, Halifax, Canada; Anwar M Batieha: Jordan University of Science and Technology, Irbid, Jordan; Rosangela L Batista: Federal University of Maranhão, São Luís, Brazil; Zhamilya Battakova: National Center of Public Healthcare, Nur-Sultan, Kazakhstan; Assembekov Batyrbek: Al-Farabi Kazakh National University, Almaty, Kazakhstan; Louise A Baur: University of Sydney, Sydney, Australia; Robert Beaglehole: University of Auckland, Auckland, New Zealand; Silvia Bel-Serrat: University College Dublin, Dublin, Ireland; Antonisamy Belavendra: Christian Medical College, Vellore, India; Habiba Ben Romdhane: University Tunis El Manar, Tunis, Tunisia; Judith Benedics: Federal Ministry of Social Affairs, Health, Care and Consumer Protection, Vienna, Austria; Mikhail Benet: Cafam University Foundation, Bogota, Colombia; Ingunn Holden Bergh: Norwegian Institute of Public Health, Oslo, Norway; Salim Berkinbayev: Kazakh National Medical University, Almaty, Kazakhstan; Antonio Bernabe-Ortiz: Universidad Peruana Cayetano Heredia, Lima, Peru; Gailute Bernotiene: Lithuanian University of Health Sciences, Kaunas, Lithuania; Heloísa Bettiol: University of São Paulo, São Paulo, Brazil; Jorge Bezerra: University of Pernambuco, Recife, Brazil; Aroor Bhagyalaxmi: B J Medical College, Ahmedabad, India; Sumit Bharadwaj: Chirayu Medical College, New Delhi, India; Santosh K Bhargava: Sunder Lal Jain Hospital, Delhi, India; Zulfiqar A Bhutta: The Hospital for Sick Children, Toronto, Canada; Aga Khan University, Karachi, Pakistan; Hongsheng Bi: Shandong University of Traditional Chinese Medicine, Jinan, China; Yufang Bi: Shanghai Jiao-Tong University School of Medicine, Shanghai, China; Daniel Bia: Universidad de la República, Montevideo, Uruguay; Elysée Claude Bika Lele: Institute of Medical Research and Medicinal Plant Studies, Yaoundé, Cameroon; Mukharram M Bikbov: Ufa Eye Research Institute, Ufa, Russian Federation; Bihungum Bista: Nepal Health Research Council, Kathmandu, Nepal; Dusko J Bjelica: University of Montenegro, Niksic, Montenegro; Peter Bjerregaard: University of Southern Denmark, Copenhagen, Denmark; Espen Bjertness: University of Oslo, Oslo, Norway; Marius B Bjertness: University of Oslo, Oslo, Norway; Cecilia Björkelund: University of Gothenburg, Gothenburg, Sweden; Katia V Bloch: Universidade Federal do Rio de Janeiro, Rio de Janeiro, Brazil; Anneke Blokstra: National Institute for Public Health and the Environment, Bilthoven, Netherlands; Simona Bo: University of Turin, Turin, Italy; Martin Bobak: University College London, London, United Kingdom; Lynne M Boddy: Liverpool John Moores University, Liverpool, United Kingdom; Bernhard O Boehm: Nanyang Technological University Singapore, Singapore, Singapore; Heiner Boeing: German Institute of Human Nutrition, Potsdam, Germany; Jose G Boggia: Universidad de la República, Montevideo, Uruguay; Elena Bogova: Endocrinology Research Centre, Moscow, Russian Federation; Carlos P Boissonnet: Centro de Educación Médica e Investigaciones Clínicas, Buenos Aires, Argentina; Stig E Bojesen: Copenhagen University Hospital, Copenhagen, Denmark; University of Copenhagen, Copenhagen, Denmark; Marialaura Bonaccio: IRCCS Neuromed, Pozzilli, Italy; Vanina Bongard: Toulouse University School of Medicine, Toulouse, France; Alice

Bonilla-Vargas: Caja Costarricense de Seguro Social, San José, Costa Rica; Matthias Bopp: University of Zurich, Zurich, Switzerland; Herman Borghs: University Hospital KU Leuven, Leuven, Belgium; Lien Braeckevelt: Flemish Agency for Care and Health, Brussels, Belgium; Lutgart Braeckman: Ghent University, Ghent, Belgium; Marjolijn CE Bragt: FrieslandCampina, Amersfoort, Netherlands; Imperia Brajkovich: Universidad Central de Venezuela, Caracas, Venezuela; Francesco Branca: World Health Organization, Geneva, Switzerland; Juergen Breckenkamp: Bielefeld University, Bielefeld, Germany; João Breda: World Health Organization Regional Office for Europe, Moscow, Russian Federation; Hermann Brenner: German Cancer Research Center, Heidelberg, Germany; Lizzy M Brewster: University of Amsterdam, Amsterdam, Netherlands; Garry R Brian: The Fred Hollows Foundation, Auckland, New Zealand; Lacramioara Brinduse: University of Medicine and Pharmacy Bucharest, Bucharest, Romania; Sinead Brophy: Swansea University, Swansea, United Kingdom; Graziella Bruno: University of Turin, Turin, Italy; H Bas Bueno-de-Mesquita: National Institute for Public Health and the Environment, Bilthoven, Netherlands; Anna Bugge: University College Copenhagen, Copenhagen, Denmark; Marta Buoncristiano: World Health Organization Regional Office for Europe, Moscow, Russian Federation; Genc Burazeri: Institute of Public Health, Tirana, Albania; Con Burns: Munster Technological University, Cork, Ireland; Antonio Cabrera de León: Universidad de La Laguna, Tenerife, Spain; Joseph Cacciottolo: University of Malta, Msida, Malta; Hui Cai: Vanderbilt University, Nashville, United States; Tilema Cama: Ministry of Health, Tongatapu, Tonga; Christine Cameron: Canadian Fitness and Lifestyle Research Institute, Ottawa, Canada; José Camolas: Hospital Santa Maria, Lisbon, Portugal; Günay Can: Istanbul University - Cerrahpasa, Istanbul, Turkey; Ana Paula C Cândido: Universidade Federal de Juiz de Fora, Juiz de Fora, Brazil; Felicia Cañete: Ministry of Public Health, Asunción, Paraguay; Mario V Capanzana: Food and Nutrition Research Institute, Taguig, Philippines; Nadežda Capková: National Institute of Public Health, Prague, Czech Republic; Eduardo Capuano: Gaetano Fucito Hospital, Mercato San Severino, Italy; Vincenzo Capuano: Gaetano Fucito Hospital, Mercato San Severino, Italy; Marloes Cardol: University of Groningen, Groningen, Netherlands; Viviane C Cardoso: University of São Paulo, São Paulo, Brazil; Axel C Carlsson: Karolinska Institutet, Huddinge, Sweden; Esteban Carmuega: Centro de Estudios sobre Nutrición Infantil, Buenos Aires, Argentina; Joana Carvalho: University of Porto, Porto, Portugal; José A Casajús: University of Zaragoza, Zaragoza, Spain; Felipe F Casanueva: Santiago de Compostela University, Santiago de Compostela, Spain; Ertugrul Celikcan: Ministry of Health, Ankara, Turkey; Laura Censi: Council for Agricultural Research and Economics, Rome, Italy; Marvin Cervantes-Loaiza: Caja Costarricense de Seguro Social, San José, Costa Rica; Juraci A Cesar: Federal University of Rio Grande, Rio Grande, Brazil; Snehalatha Chamukuttan: India Diabetes Research Foundation, Chennai, India; Angelique W Chan: Duke-NUS Medical School, Singapore, Singapore; Queenie Chan: Imperial College London, London, United Kingdom; Himanshu K Chaturvedi: ICMR - National Institute of Medical Statistics, New Delhi, India; Nish Chaturvedi: University College London, London, United Kingdom; Norsyamlina Che Abdul Rahim: Ministry of Health, Kuala Lumpur, Malaysia; Miao Li Chee: Singapore Eye Research Institute, Singapore, Singapore; Chien-Jen Chen: Academia Sinica, Taipei, Taiwan; Fangfang Chen: Capital Institute of Pediatrics, Beijing, China; Huashuai Chen: Duke University, Durham, United States; Shuohua Chen: Kailuan General Hospital, Tangshan, China; Zhengming Chen: University of Oxford, Oxford, United Kingdom; Ching-Yu Cheng: Duke-NUS Medical School, Singapore, Singapore; Bahman Cheraghian: Ahvaz Jundishapur University of Medical Sciences, Ahvaz, Islamic Republic of Iran; Angela Chetrit: The Gertner Institute for Epidemiology and Health Policy Research, Ramat Gan, Israel; Ekaterina Chikova-Iscener: National Centre of Public Health and Analyses, Sofia, Bulgaria; Arnaud Chiolero: University of Fribourg, Fribourg, Switzerland; Shu-Ti Chiou: Ministry of Health and Welfare, Taipei, Taiwan; María-Dolores Chirlaque: CIBER Epidemiología y Salud Pública, Murcia, Spain; Belong Cho: Seoul National University, Seoul, Republic of Korea; Kaare Christensen: University of Southern Denmark, Odense, Denmark; Diego G Christofaro: Universidade Estadual Paulista, Presidente Prudente, Brazil; Jerzy Chudek: Medical University of Silesia, Katowice, Poland; Renata Cifkova: Charles University, Prague, Czech Republic; Thomayer Hospital, Prague, Czech Republic; Michelle Cilia: Primary Health Care, Floriana, Malta; Eliza Cinteza: Carol Davila University of Medicine and Pharmacy, Bucharest, Romania; Frank Claessens: Katholieke Universiteit Leuven, Leuven, Belgium; Janine Clarke: Statistics Canada,

Ottawa, Canada; Els Clays: Ghent University, Ghent, Belgium; Emmanuel Cohen: UMR CNRS-MNHN 7206 Eco-anthropologie, Marseille, France; Hans Concin: Agency for Preventive and Social Medicine, Bregenz, Austria; Susana C Confortin: Federal University of Maranhão, São Luís, Brazil; Cyrus Cooper: University of Southampton, Southampton, United Kingdom; Tara C Coppinger: Munster Technological University, Cork, Ireland; Eva Corpeleijn: University of Groningen, Groningen, Netherlands; Simona Costanzo: IRCCS Neuromed, Pozzilli, Italy; Dominique Cottel: Institut Pasteur de Lille, Lille, France; Chris Cowell: University of Sydney, Sydney, Australia; Cora L Craig: Canadian Fitness and Lifestyle Research Institute, Ottawa, Canada; Amelia C Crampin: Malawi Epidemiology and Intervention Research Unit, Lilongwe, Malawi; Ana B Crujeiras: CIBEROBN, Madrid, Spain; Semánová Csilla: University of Debrecen, Debrecen, Hungary; Alexandra M Cucu: University of Medicine and Pharmacy Carol Davila, Bucharest, Romania; Liufu Cui: Kailuan General Hospital, Tangshan, China; Felipe V Cureau: Universidade Federal do Rio Grande do Sul, Porto Alegre, Brazil; Ewelina Czenczek-Lewandowska: University of Rzeszów, Rzeszów, Poland; Graziella D'Arrigo: National Research Council, Reggio Calabria, Italy; Eleonora d'Orsi: Federal University of Santa Catarina, Florianópolis, Brazil; Liliana Dacica: Eftimie Murgu University Resita, Resita, Romania; María Ángeles Dal Re Saavedra: Spanish Agency for Food Safety and Nutrition, Madrid, Spain; Jean Dallongeville: Institut Pasteur de Lille, Lille, France; Camilla T Damsgaard: University of Copenhagen, Copenhagen, Denmark; Rachel Dankner: The Gertner Institute for Epidemiology and Health Policy Research, Ramat Gan, Israel; Thomas M Dantoft: Bispebjerg and Frederiksberg Hospital, Copenhagen, Denmark; Parasmani Dasgupta: Indian Statistical Institute, Kolkata, India; Saeed Dastgiri: Tabriz Health Services Management Research Center, Tabriz, Islamic Republic of Iran; Luc Dauchet: University of Lille, Lille, France; Lille University Hospital, Lille, France; Kairat Davletov: Al-Farabi Kazakh National University, Almaty, Kazakhstan; Guy De Backer: Ghent University, Ghent, Belgium; Dirk De Bacquer: Ghent University, Ghent, Belgium; Giovanni de Gaetano: IRCCS Neuromed, Pozzilli, Italy; Stefaan De Henauw: Ghent University, Ghent, Belgium; Paula Duarte de Oliveira: Federal University of Pelotas, Pelotas, Brazil; David De Ridder: Geneva University Hospitals, Geneva, Switzerland; Karin De Ridder: Sciensano, Brussels, Belgium; Susanne R de Rooij: University Medical Centers, Groningen, Netherlands; University of Amsterdam, Amsterdam, Netherlands; Delphine De Smedt: Ghent University, Ghent, Belgium; Mohan Deepa: Madras Diabetes Research Foundation, Chennai, India; Alexander D Deev: National Research Centre for Preventive Medicine, Moscow, Russian Federation; Vincent Jr DeGennaro: Innovating Health International, Port-au-Prince, Haiti; Abbas Dehghan: Imperial College London, London, United Kingdom; Hélène Delisle: University of Montreal, Montreal, Canada; Francis Delpeuch: French National Research Institute for Sustainable Development, Montpellier, France; Stefaan Demarest: Sciensano, Brussels, Belgium; Elaine Dennison: University of Southampton, Southampton, United Kingdom; Katarzyna Dereń: University of Rzeszów, Rzeszów, Poland; Valérie Deschamps: French Public Health Agency, St Maurice, France; Meghnath Dhimal: Nepal Health Research Council, Kathmandu, Nepal; Augusto F Di Castelnuovo: Mediterranea Cardiocentro, Naples, Italy; Juvenal Soares Dias-da-Costa: Universidade do Vale do Rio dos Sinos, São Leopoldo, Brazil; María Elena Díaz-Sánchez: National Institute of Hygiene, Epidemiology and Microbiology, Havana, Cuba; Alejandro Diaz: National Council of Scientific and Technical Research, Buenos Aires, Argentina; Zivka Dika: University of Zagreb, Zagreb, Croatia; Shirin Djalalinia: Ministry of Health and Medical Education, Tehran, Islamic Republic of Iran; Visnja Djordjic: University of Novi Sad, Novi Sad, Serbia; Ha TP Do: National Institute of Nutrition, Hanoi, Viet Nam; Annette J Dobson: University of Queensland, Brisbane, Australia; Maria Benedetta Donati: IRCCS Neuromed, Pozzilli, Italy; Chiara Donfrancesco: Istituto Superiore di Sanità, Rome, Italy; Silvana P Donoso: Universidad de Cuenca, Cuenca, Ecuador; Angela Döring: Helmholtz Zentrum München, Munich, Germany; Maria Dorobantu: Carol Davila University of Medicine and Pharmacy, Bucharest, Romania; Ahmad Reza Dorosty: Tehran University of Medical Sciences, Tehran, Islamic Republic of Iran; Kouamelan Doua: Ministère de la Santé et de l'Hygiène Publique, Abidjan, Côte d'Ivoire; Nico Dragano: University Hospital Düsseldorf, Düsseldorf, Germany; Wojciech Drygas: National Institute of Cardiology, Warsaw, Poland; Medical University of Lodz, Lodz, Poland; Jia Li Duan: Beijing Center for Disease Prevention and Control, Beijing, China; Charmaine A Duante: Food and Nutrition Research Institute, Taguig, Philippines; Priscilla Duboz: UMI 3189 ESS, Marseille,

France; Rosemary B Duda: Beth Israel Deaconess Medical Center, Boston, United States; Harvard Medical School, Boston, United States; Vesselka Duleva: National Centre of Public Health and Analyses, Sofia, Bulgaria; Virginija Dulskiene: Lithuanian University of Health Sciences, Kaunas, Lithuania; Samuel C Dumith: Federal University of Rio Grande, Rio Grande, Brazil; Anar Dushpanova: Scuola Superiore Sant'Anna, Pisa, Italy; Al-Farabi Kazakh National University, Almaty, Kazakhstan; Vilnis Dzerve: University of Latvia, Riga, Latvia; Elzbieta Dziankowska-Zaborszczyk: Medical University of Lodz, Lodz, Poland; Ricky Eddie: Ministry of Health and Medical Services, Gizo, Solomon Islands; Ebrahim Eftekhar: Hormozgan University of Medical Sciences, Bandar Abbas, Islamic Republic of Iran; Eruke E Egbagbe: University of Benin, Benin City, Nigeria; Robert Eggertsen: University of Gothenburg, Gothenburg, Sweden; Sareh Eghtesad: Tehran University of Medical Sciences, Tehran, Islamic Republic of Iran; Gabriele Eiben: University of Skövde, Skövde, Sweden; Ulf Ekelund: Norwegian School of Sport Sciences, Oslo, Norway; Mohammad El-Khateeb: National Center for Diabetes, Endocrinology and Genetics, Amman, Jordan; Jalila El Ati: National Institute of Nutrition and Food Technology, Tunis, Tunisia; Denise Eldemire-Shearer: The University of the West Indies, Kingston, Jamaica; Marie Eliasen: Bispebjerg and Frederiksberg Hospital, Copenhagen, Denmark; Paul Elliott: Imperial College London, London, United Kingdom; Reina Engle-Stone: University of California Davis, Davis, United States; Macia Enguerran: UMI 3189 ESS, Marseille, France; Rajiv T Erasmus: University of Stellenbosch, Cape Town, South Africa; Raimund Erbel: University of Duisburg-Essen, Duisburg, Germany; Cihangir Erem: Karadeniz Technical University, Trabzon, Turkey; Louise Eriksen: University of Southern Denmark, Copenhagen, Denmark; Johan G Eriksson: University of Helsinki, Helsinki, Finland; Jorge Escobedo-de la Peña: Instituto Mexicano del Seguro Social, Mexico City, Mexico; Saeid Eslami: Mashhad University of Medical Sciences, Mashhad, Islamic Republic of Iran; Ali Esmaeili: Rafsanjan University of Medical Sciences, Rafsanjan, Islamic Republic of Iran; Alun Evans: Queen's University of Belfast, Belfast, United Kingdom; David Faeh: University of Zurich, Zurich, Switzerland; Albina A Fakhretdinova: Ufa Eye Research Institute, Ufa, Russian Federation; Caroline H Fall: University of Southampton, Southampton, United Kingdom; Elnaz Faramarzi: Tabriz University of Medical Sciences, Tabriz, Islamic Republic of Iran; Mojtaba Farjam: Fasa University of Medical Sciences, Fasa, Islamic Republic of Iran; Victoria Farrugia Sant'Angelo: Primary Health Care, Floriana, Malta; Farshad Farzadfar: Non-Communicable Diseases Research Center, Tehran, Islamic Republic of Iran; Mohammad Reza Fattahi: Shiraz University of Medical Sciences, Shiraz, Islamic Republic of Iran; Asher Fawwad: Baqai Medical University, Karachi, Pakistan; Francisco J Felix-Redondo: Centro de Salud Villanueva Norte, Badajoz, Spain; Trevor S Ferguson: The University of the West Indies, Kingston, Jamaica; Romulo A Fernandes: Universidade Estadual Paulista, Presidente Prudente, Brazil; Daniel Fernández-Bergés: Hospital Don Benito-Villanueva de la Serena, Badajoz, Spain; Daniel Ferrante: Ministry of Health, Buenos Aires, Argentina; Thomas Ferrao: Statistics Canada, Ottawa, Canada; Marika Ferrari: Council for Agricultural Research and Economics, Rome, Italy; Marco M Ferrario: University of Insubria, Varese, Italy; Catterina Ferreccio: Pontificia Universidad Católica de Chile, Santiago, Chile; Eldridge Ferrer: Food and Nutrition Research Institute, Taguig, Philippines; Jean Ferrieres: Toulouse University School of Medicine, Toulouse, France; Thamara Hubler Figueiró: Federal University of Santa Catarina, Florianópolis, Brazil; Anna Fijalkowska: Institute of Mother and Child, Warsaw, Poland; Günther Fink: Swiss Tropical and Public Health Institute, Basel, Switzerland; University of Basel, Basel, Switzerland; Krista Fischer: University of Tartu, Tartu, Estonia; Leng Huat Foo: Universiti Sains Malaysia, Kelantan, Malaysia; Maria Forsner: Umeå University, Umeå, Sweden; Heba M Fouad: World Health Organization Regional Office for the Eastern Mediterranean, Cairo, Egypt; Damian K Francis: The University of the West Indies, Kingston, Jamaica; Maria do Carmo Franco: Federal University of São Paulo, São Paulo, Brazil; Ruth Frikke-Schmidt: University of Copenhagen, Copenhagen, Denmark; Copenhagen University Hospital, Copenhagen, Denmark; Guillermo Frontera: Hospital Universitario Son Espases, Palma, Spain; Flavio D Fuchs: Hospital de Clinicas de Porto Alegre, Porto Alegre, Brazil; Sandra C Fuchs: Universidade Federal do Rio Grande do Sul, Porto Alegre, Brazil; Isti I Fujiati: Universitas Sumatera Utara, Medan, Indonesia; Yuki Fujita: Kindai University, Osaka-Sayama, Japan; Matsuda Fumihiko: Kyoto University, Kyoto, Japan; Takuro Furusawa: Kyoto University, Kyoto, Japan; Zbigniew Gaciong: Medical University of Warsaw, Warsaw, Poland; Mihai Gafencu: Victor Babes University of Medicine and Pharmacy Timisoara, Timisoara,

Romania; Andrzej Galbarczyk: Jagiellonian University Medical College, Kraków, Poland; Henrike Galenkamp: University of Amsterdam, Amsterdam, Netherlands; Daniela Galeone: Ministero della Salute DG Prevenzione Sanitaria, Rome, Italy; Myriam Galfo: Council for Agricultural Research and Economics, Rome, Italy; Fabio Galvano: University of Catania, Catania, Italy; Jingli Gao: Kailuan General Hospital, Tangshan, China; Manoli Garcia-de-la-Hera: CIBER en Epidemiología y Salud Pública, Alicante, Spain; Marta García-Solano: Spanish Agency for Food Safety and Nutrition, Madrid, Spain; Dickman Gareta: Africa Health Research Institute, Mtubatuba, South Africa; Sarah P Garnett: University of Sydney, Sydney, Australia; Jean-Michel Gaspoz: Geneva University Medical School, Geneva, Switzerland; Magda Gasull: CIBER en Epidemiología y Salud Pública, Barcelona, Spain; Adroaldo Cesar Araujo Gaya: Universidade Federal do Rio Grande do Sul, Porto Alegre, Brazil; Anelise Reis Gaya: Universidade Federal do Rio Grande do Sul, Porto Alegre, Brazil; Andrea Gazzinelli: Universidade Federal de Minas Gerais, Belo Horizonte, Brazil; Ulrike Gehring: Utrecht University, Utrecht, Netherlands; Harald Geiger: Agency for Preventive and Social Medicine, Bregenz, Austria; Johanna M Geleijnse: Wageningen University, Wageningen, Netherlands; Ali Ghanbari: Non-Communicable Diseases Research Center, Tehran, Islamic Republic of Iran; Erfan Ghasemi: Non-Communicable Diseases Research Center, Tehran, Islamic Republic of Iran; Oana-Florentina Gheorghe-Fronea: Carol Davila University of Medicine and Pharmacy, Bucharest, Romania; Simona Giampaoli: Istituto Superiore di Sanità, Rome, Italy; Francesco Gianfagna: University of Insubria, Varese, Italy; Mediterranea Cardiocentro, Naples, Italy; Tiffany K Gill: University of Adelaide, Adelaide, Australia; Jonathan Giovannelli: University of Lille, Lille, France; Lille University Hospital, Lille, France; Glen Gironella: Food and Nutrition Research Institute, Taguig, Philippines; Aleksander Giwercman: Lund University, Lund, Sweden; Konstantinos Gkiouras: Aristotle University of Thessaloniki, Thessaloniki, Greece; Justyna Godos: University of Catania, Catania, Italy; Sibel Gogen: Ministry of Health, Ankara, Turkey; Marcel Goldberg: Institut National de la Santé et de la Recherche Médicale, Villejuif, France; Paris University, Paris, France; Rebecca A Goldsmith: Ministry of Health, Jerusalem, Israel; David Goltzman: McGill University, Montreal, Canada; Santiago F Gómez: Gasol Foundation, Barcelona, Spain; University of Lleida, Sant Boi de Llobregat, Spain; Aleksandra Gomula: PASs Hirszfeld Institute of Immunology and Experimental Therapy, Wroclaw, Poland; Bruna Goncalves Cordeiro da Silva: Federal University of Pelotas, Pelotas, Brazil; Helen Gonçalves: Federal University of Pelotas, Pelotas, Brazil; David A Gonzalez-Chica: University of Adelaide, Adelaide, Australia; Marcela Gonzalez-Gross: Universidad Politécnica de Madrid, Madrid, Spain; Margot González-Leon: Instituto Mexicano del Seguro Social, Mexico City, Mexico; Juan P González-Rivas: St Anne's University Hospital, Brno, Czech Republic; Clicerio González-Villalpando: National Institute of Public Health, Cuernavaca, Mexico; María-Elena González-Villalpando: Centro de Estudios en Diabetes A.C., Mexico City, Mexico; Angel R Gonzalez: Universidad Autónoma de Santo Domingo, Santo Domingo, Dominican Republic; Frederic Gottrand: University of Lille, Lille, France; Antonio Pedro Graça: Ministry of Health, Lisbon, Portugal; Sidsel Graff-Iversen: Norwegian Institute of Public Health, Oslo, Norway; Dušan Grafnetter: Institute for Clinical and Experimental Medicine, Prague, Czech Republic; Aneta Grajda: Children's Memorial Health Institute, Warsaw, Poland; Maria G Grammatikopoulou: Aristotle University of Thessaloniki, Thessaloniki, Greece; Ronald D Gregor: Dalhousie University, Halifax, Canada; Tomasz Grodzicki: Jagiellonian University Medical College, Kraków, Poland; Else Karin Grøholt: Norwegian Institute of Public Health, Oslo, Norway; Anders Grøntved: University of Southern Denmark, Odense, Denmark; Giuseppe Grosso: University of Catania, Catania, Italy; Gabriella Gruden: University of Turin, Turin, Italy; Dongfeng Gu: National Center of Cardiovascular Diseases, Beijing, China; Emanuela Gualdi-Russo: University of Ferrara, Ferrara, Italy; Pilar Guallar-Castillón: Universidad Autónoma de Madrid CIBERESP, Madrid, Spain; Andrea Gualtieri: Authority Sanitaria San Marino, San Marino, San Marino; Elias F Gudmundsson: Icelandic Heart Association, Kopavogur, Iceland; Vilmundur Gudnason: University of Iceland, Reykjavik, Iceland; Ramiro Guerrero: Universidad Icesi, Cali, Colombia; Idris Guessous: Geneva University Hospitals, Geneva, Switzerland; Andre L Guimaraes: State University of Montes Claros, Montes Claros, Brazil; Martin C Gulliford: King's College London, London, United Kingdom; Johanna Gunnlaugsdottir: Icelandic Heart Association, Kopavogur, Iceland; Marc J Gunter: International Agency for Research on Cancer, Lyon, France; Xiu-Hua Guo: Capital Medical University, Beijing, China; Yin Guo: Capital Medical

University Beijing Tongren Hospital, Beijing, China; Prakash C Gupta: Healis-Sekhsaria Institute for Public Health, Navi Mumbai, India; Rajeev Gupta: Eternal Heart Care Centre and Research Institute, Jaipur, India; Oye Gureje: University of Ibadan, Ibadan, Nigeria; Beata Gurzkowska: Children's Memorial Health Institute, Warsaw, Poland; Enrique Gutiérrez-González: Spanish Agency for Food Safety and Nutrition, Madrid, Spain; Laura Gutierrez: Institute for Clinical Effectiveness and Health Policy, Buenos Aires, Argentina; Felix Gutzwiller: University of Zurich, Zurich, Switzerland; Seongjun Ha: National Health Insurance Service, Wonju, Republic of Korea; Farzad Hadaegh: Prevention of Metabolic Disorders Research Center, Tehran, Islamic Republic of Iran; Charalambos A Hadjigeorgiou: Research and Education Institute of Child Health, Nicosia, Cyprus; Rosa Haghshenas: Non-Communicable Diseases Research Center, Tehran, Islamic Republic of Iran; Hamid Hakimi: Rafsanjan University of Medical Sciences, Rafsanjan, Islamic Republic of Iran; Jytte Halkjær: Danish Cancer Society Research Center, Copenhagen, Denmark; Ian R Hambleton: The University of the West Indies, Cave Hill, Barbados; Behrooz Hamzeh: Kermanshah University of Medical Sciences, Kermanshah, Islamic Republic of Iran; Dominique Hange: University of Gothenburg, Gothenburg, Sweden; Abu AM Hanif: BRAC James P Grant School of Public Health, Dhaka, Bangladesh; Sari Hantunen: University of Eastern Finland, Kuopio, Finland; Jie Hao: Beijing Institute of Ophthalmology, Beijing, China; Rachakulla Hari Kumar: ICMR - National Institute of Nutrition, Hyderabad, India; Seyed Mohammad Hashemi-Shahri: Zahedan University of Medical Sciences, Zahedan, Islamic Republic of Iran; Maria Hassapidou: International Hellenic University, Thessaloniki, Greece; Jun Hata: Kyushu University, Fukuoka, Japan; Teresa Haugsgjerd: University of Bergen, Bergen, Norway; Jiang He: Tulane University, New Orleans, United States; Yuan He: National Research Institute for Health and Family Planning, Beijing, China; Yuna He: Chinese Center for Disease Control and Prevention, Beijing, China; Regina Heidinger-Felso: University of Pécs, Pécs, Hungary; Mirjam Heinen: University College Dublin, Dublin, Ireland; Tatjana Hejgaard: Danish Health Authority, Copenhagen, Denmark; Marleen Elisabeth Hendriks: Joep Lange Institute, Amsterdam, Netherlands; Rafael dos Santos Henrique: Federal University of Pernambuco, Recife, Brazil; Ana Henriques: Institute of Public Health of the University of Porto, Porto, Portugal; Leticia Hernandez Cadena: National Institute of Public Health, Cuernavaca, Mexico; Sauli Herrala: Oulu University Hospital, Oulu, Finland; Victor M Herrera: Universidad Autónoma de Bucaramanga, Bucaramanga, Colombia; Isabelle Herter-Aeberli: ETH Zurich, Zurich, Switzerland; Ramin Heshmat: Chronic Diseases Research Center, Tehran, Islamic Republic of Iran; Allan G Hill: University of Southampton, Southampton, United Kingdom; Sai Yin Ho: University of Hong Kong, Hong Kong, China; Suzanne C Ho: The Chinese University of Hong Kong, Hong Kong, China; Michael Hobbs: University of Western Australia, Perth, Australia; Michelle Holdsworth: French National Research Institute for Sustainable Development, Montpellier, France; Reza Homayounfar: Fasa University of Medical Sciences, Fasa, Islamic Republic of Iran; Clara Homs: Gasol Foundation, Spain; University Ramon Llull, Sant Boi de Llobregat, Spain; Wilma M Hopman: Kingston Health Sciences Centre, Kingston, Canada; Andrea RVR Horimoto: University of São Paulo, São Paulo, Brazil; Claudia M Hormiga: Fundación Oftalmológica de Santander, Bucaramanga, Colombia; Bernardo L Horta: Federal University of Pelotas, Pelotas, Brazil; Leila Houti: University Oran 1, Oran, Algeria; Christina Howitt: The University of the West Indies, Cave Hill, Barbados; Thein Thein Htay: Independent Public Health Specialist, Nay Pyi Taw, Myanmar; Aung Soe Htet: Ministry of Health and Sports, Nay Pyi Taw, Myanmar; Maung Maung Than Htike: Ministry of Health and Sports, Nay Pyi Taw, Myanmar; Yonghua Hu: Peking University, Beijing, China; José María Huerta: CIBER en Epidemiología y Salud Pública, Murcia, Spain; Ilpo Tapani Huhtaniemi: Imperial College London, London, United Kingdom; Laetitia Huiart: Luxembourg Institute of Health, Luxembourg, Luxembourg; Constanta Huidumac Petrescu: National Institute of Public Health, Bucharest, Romania; Martijn Huisman: VU University Medical Center, Amsterdam, Netherlands; Abdullatif Husseini: Birzeit University, Birzeit, State of Palestine; Chinh Nguyen Huu: National Institute of Nutrition, Hanoi, Viet Nam; Inge Huybrechts: International Agency for Research on Cancer, Lyon, France; Nahla Hwalla: American University of Beirut, Beirut, Lebanon; Jolanda Hyska: Institute of Public Health, Tirana, Albania; Licia Iacoviello: IRCCS Neuromed, Pozzilli, Italy; University of Insubria, Varese, Italy; Jesús M Ibarluzea: CIBER en Epidemiología y Salud Pública, San Sebastian, Spain; Mohsen M Ibrahim: Cairo University, Cairo, Egypt; Norazizah Ibrahim Wong: Ministry of Health, Kuala Lumpur,

Malaysia; M Arfan Ikram: Erasmus Medical Center Rotterdam, Rotterdam, Netherlands; Violeta Iotova: Medical University Varna, Varna, Bulgaria; Vilma E Irazola: Institute for Clinical Effectiveness and Health Policy, Buenos Aires, Argentina; Takafumi Ishida: The University of Tokyo, Tokyo, Japan; Muhammad Islam: The Hospital for Sick Children, Toronto, Canada; Sheikh Mohammed Shariful Islam: Deakin University, Geelong, Australia; Masanori Iwasaki: Tokyo Metropolitan Institute of Gerontology, Tokyo, Japan; Jeremy M Jacobs: Hadassah University Medical Center, Jerusalem, Israel; Hashem Y Jaddou: Jordan University of Science and Technology, Irbid, Jordan; Tazeen Jafar: Duke-NUS Medical School, Singapore, Singapore; Kenneth James: The University of the West Indies, Kingston, Jamaica; Kazi M Jamil: Kuwait Institute for Scientific Research, Safat, Kuwait; Konrad Jamrozik: University of Adelaide, Adelaide, Australia; Imre Janszky: Norwegian University of Science and Technology, Trondheim, Norway; Edward Janus: University of Melbourne, Melbourne, Australia; Juel Jarani: Sports University of Tirana, Tirana, Albania; Marjo-Riitta Jarvelin: Imperial College London, London, United Kingdom; University of Oulu, Oulu, Finland; Grazyna Jasienska: Jagiellonian University Medical College, Kraków, Poland; Ana Jelakovic: University Hospital Center Zagreb, Zagreb, Croatia; Bojan Jelakovic: University of Zagreb School of Medicine, Zagreb, Croatia; Garry Jennings: Heart Foundation, Melbourne, Australia; Anjani Kumar Jha: Nepal Health Research Council, Kathmandu, Nepal; Chao Qiang Jiang: Guangzhou 12th Hospital, Guangzhou, China; Ramon O Jimenez: Universidad Eugenio Maria de Hostos, Santo Domingo, Dominican Republic; Karl-Heinz Jöckel: University of Duisburg-Essen, Duisburg, Germany; Michel Joffres: Simon Fraser University, Burnaby, Canada; Mattias Johansson: International Agency for Research on Cancer, Lyon, France; Jari J Jokelainen: Oulu University Hospital, Oulu, Finland; Jost B Jonas: Institute of Molecular and Clinical Ophthalmology Basel, Basel, Switzerland; Jitendra Jonnagaddala: University of New South Wales, Sydney, Australia; Torben Jørgensen: Bispebjerg and Frederiksberg Hospital, Copenhagen, Denmark; Pradeep Joshi: World Health Organization Country Office, Delhi, India; Farahnaz Joukar: Guilan University of Medical Sciences, Rasht, Islamic Republic of Iran; Dragana P Jovic: Institute of Public Health, Belgrade, Serbia; Jacek J Jóźwiak: University of Opole, Opole, Poland; Anne Juolevi: Finnish Institute for Health and Welfare, Helsinki, Finland; Gregor Jurak: University of Ljubljana, Ljubljana, Slovenia; Iulia Jurca Simina: Victor Babes University of Medicine and Pharmacy Timisoara, Timisoara, Romania; Vesna Juresa: University of Zagreb, Zagreb, Croatia; Rudolf Kaaks: German Cancer Research Center, Heidelberg, Germany; Felix O Kaducu: Gulu University, Gulu, Uganda; Anthony Kafatos: University of Crete, Heraklion, Greece; Eero O Kajantie: Finnish Institute for Health and Welfare, Helsinki, Finland; Zhanna Kalmatayeva: Al-Farabi Kazakh National University, Almaty, Kazakhstan; Ofra Kalter-Leibovici: The Gertner Institute for Epidemiology and Health Policy Research, Ramat Gan, Israel; Yves Kameli: French National Research Institute for Sustainable Development, Montpellier, France; Freja B Kampmann: Bispebjerg and Frederiksberg Hospital, Copenhagen, Denmark; Kodanda R Kanala: Sri Venkateswara University, Tirupati, India; Srinivasan Kannan: Sree Chitra Tirunal Institute for Medical Sciences and Technology, Trivandrum, India; Efthymios Kapantais: Hellenic Medical Association for Obesity, Athens, Greece; Argyro Karakosta: National and Kapodistrian University of Athens, Athens, Greece; Line L Kårhus: Bispebjerg and Frederiksberg Hospital, Copenhagen, Denmark; Khem B Karki: Maharajgunj Medical Campus, Kathmandu, Nepal; Marzieh Katibeh: Aarhus University, Aarhus, Denmark; Joanne Katz: Johns Hopkins Bloomberg School of Public Health, Baltimore, United States; Peter T Katzmarzyk: Pennington Biomedical Research Center, Baton Rouge, United States; Jussi Kauhanen: University of Eastern Finland, Kuopio, Finland; Prabhdeep Kaur: National Institute of Epidemiology, Chennai, India; Maryam Kavousi: Erasmus Medical Center Rotterdam, Rotterdam, Netherlands; Gyulli M Kazakbaeva: Ufa Eye Research Institute, Ufa, Russian Federation; Ulrich Keil: University of Münster, Münster, Germany; Lital Keinan Boker: Israel Center for Disease Control, Ramat Gan, Israel; Sirkka Keinänen-Kiukaanniemi: Oulu University Hospital, Oulu, Finland; Roya Kelishadi: Research Institute for Primordial Prevention of Non-communicable Disease, Isfahan, Islamic Republic of Iran; Cecily Kelleher: University College Dublin, Dublin, Ireland; Han CG Kemper: Amsterdam UMC Public Health Research Institute, Amsterdam, Netherlands; Andre P Kengne: South African Medical Research Council, Cape Town, South Africa; Maryam Keramati: Mashhad University of Medical Sciences, Mashhad, Islamic Republic of Iran; Alina Kerimkulova: Kyrgyz State Medical Academy, Bishkek, Kyrgyzstan;

Mathilde Kersting: Research Institute of Child Nutrition, Dortmund, Germany; Timothy Key: University of Oxford, Oxford, United Kingdom; Yousef Saleh Khader: Jordan University of Science and Technology, Irbid, Jordan; Davood Khalili: Shahid Beheshti University of Medical Sciences, Tehran, Islamic Republic of Iran; Kay-Tee Khaw: University of Cambridge, Cambridge, United Kingdom; Bahareh Kheiri: Shahid Beheshti University of Medical Sciences, Tehran, Islamic Republic of Iran; Motahareh Kheradmand: Mazandaran University of Medical Sciences, Sari, Islamic Republic of Iran; Alireza Khosravi: Hypertension Research Center, Isfahan, Islamic Republic of Iran; Ilse MSL Khouw: FrieslandCampina, Amersfoort, Netherlands; Ursula Kiechl-Kohlendorfer: Medical University of Innsbruck, Innsbruck, Austria; Stefan Kiechl: Medical University of Innsbruck, Innsbruck, Austria; VASCage, Innsbruck, Austria; Japhet Killewo: Muhimbili University of Health and Allied Sciences, Dar es Salaam, United Republic of Tanzania; Dong Wook Kim: National Health Insurance Service, Wonju, Republic of Korea; Hyeon Chang Kim: Yonsei University College of Medicine, Seoul, Republic of Korea; Jeongseon Kim: National Cancer Center, Goyang-si, Republic of Korea; Jenny M Kindblom: University of Gothenburg, Gothenburg, Sweden; Sahlgrenska University Hospital, Gothenburg, Sweden; Heidi Klakk: University College South Denmark, Haderslev, Denmark; Magdalena Klimek: Jagiellonian University Medical College, Kraków, Poland; Jeannette Klimont: Statistics Austria, Vienna, Austria; Jurate Klumbiene: Lithuanian University of Health Sciences, Kaunas, Lithuania; Michael Knoflach: Medical University of Innsbruck, Innsbruck, Austria; Bhawesh Koirala: B P Koirala Institute of Health Sciences, Dharan, Nepal; Elin Kolle: Norwegian School of Sport Sciences, Oslo, Norway; Patrick Kolsteren: Ghent University, Ghent, Belgium; Jürgen König: University of Vienna, Vienna, Austria; Raija Korpelainen: University of Oulu, Oulu, Finland; Oulu Deaconess Institute Foundation, Oulu, Finland; Paul Korrovits: Tartu University Clinics, Tartu, Estonia; Magdalena Korzycka: Institute of Mother and Child, Warsaw, Poland; Jelena Kos: University Hospital Center Zagreb, Zagreb, Croatia; Seppo Koskinen: Finnish Institute for Health and Welfare, Helsinki, Finland; Katsuyasu Kouda: Kansai Medical University, Hirakata, Japan; Viktoria A Kovacs: Hungarian School Sport Federation, Budapest, Hungary; Sudhir Kowlessur: Ministry of Health and Quality of Life, Port Louis, Mauritius; Slawomir Koziel: PASs Hirszfeld Institute of Immunology and Experimental Therapy, Wroclaw, Poland; Jana Kratenova: National Institute of Public Health, Prague, Czech Republic; Wolfgang Kratzer: University Hospital Ulm, Ulm, Germany; Susi Kriemler: University of Zurich, Zurich, Switzerland; Peter Lund Kristensen: University of Southern Denmark, Odense, Denmark; Steinar Krokstad: Norwegian University of Science and Technology, Trondheim, Norway; Daan Kromhout: University of Groningen, Groningen, Netherlands; Herculina S Kruger: North-West University, Potchefstroom, South Africa; Ruzena Kubinova: National Institute of Public Health, Prague, Czech Republic; Renata Kuciene: Lithuanian University of Health Sciences, Kaunas, Lithuania; Urho M Kujala: University of Jyväskylä, Jyväskylä, Finland; Enisa Kujundzic: Institute of Public Health, Podgorica, Montenegro; Zbigniew Kulaga: Children's Memorial Health Institute, Warsaw, Poland; R Krishna Kumar: Amrita Institute of Medical Sciences, Cochin, India; Marie Kunešová: Institute of Endocrinology, Prague, Czech Republic; Pawel Kurjata: National Institute of Cardiology, Warsaw, Poland; Yadlapalli S Kusuma: All India Institute of Medical Sciences, New Delhi, India; Kari Kuulasmaa: Finnish Institute for Health and Welfare, Helsinki, Finland; Catherine Kyobutungi: African Population and Health Research Center, Nairobi, Kenya; Quang Ngoc La: Hanoi University of Public Health, Hanoi, Viet Nam; Fatima Zahra Laamiri: Hassan First University of Settat, Settat, Morocco; Tiina Laatikainen: Finnish Institute for Health and Welfare, Helsinki, Finland; Carl Lachat: Ghent University, Ghent, Belgium; Youcef Laid: Ministry of Health, Algiers, Algeria; Tai Hing Lam: University of Hong Kong, Hong Kong, China; Christina-Paulina Lambrinou: Harokopio University, Athens, Greece; Edwige Landais: French National Research Institute for Sustainable Development, Montpellier, France; Vera Lanska: Institute for Clinical and Experimental Medicine, Prague, Czech Republic; Georg Lappas: Sahlgrenska Academy, Gothenburg, Sweden; Bagher Larijani: Endocrinology and Metabolism Research Center, Tehran, Islamic Republic of Iran; Tint Swe Latt: University of Public Health, Yangon, Myanmar; Laura Lauria: Istituto Superiore di Sanità, Rome, Italy; Maria Lazo-Porras: Universidad Peruana Cayetano Heredia, Lima, Peru; Gwenaëlle Le Coroller: Luxembourg Institute of Health, Strassen, Luxembourg; Khanh Le Nguyen Bao: National Institute of Nutrition, Hanoi, Viet Nam; Agnès Le Port: International Food Policy Research Institute, Dakar, Senegal; Tuyen D Le: National Institute

of Nutrition, Hanoi, Viet Nam; Jeannette Lee: National University of Singapore, Singapore, Singapore; Jeonghee Lee: National Cancer Center, Goyang-si, Republic of Korea; Paul H Lee: Hong Kong Polytechnic University, Hong Kong, China; Nils Lehmann: University of Duisburg-Essen, Duisburg, Germany; Terho Lehtimäki: Tampere University Hospital, Tampere, Finland; Tampere University, Tampere, Finland; Daniel Lemogoum: University of Douala, Douala, Cameroon; Naomi S Levitt: University of Cape Town, Cape Town, South Africa; Yanping Li: Harvard TH Chan School of Public Health, Boston, United States; Merike Liivak: National Institute for Health Development, Tallinn, Estonia; Christa L Lilly: West Virginia University, Morgantown, United States; Wei-Yen Lim: National University of Singapore, Singapore, Singapore; M Fernanda Lima-Costa: Oswaldo Cruz Foundation Rene Rachou Research Institute, Belo Horizonte, Brazil; Hsien-Ho Lin: National Taiwan University, Taipei, Taiwan; Xu Lin: University of Chinese Academy of Sciences, Shanghai, China; Yi-Ting Lin: Uppsala University, Uppsala, Sweden; Lars Lind: Uppsala University, Uppsala, Sweden; Allan Linneberg: Bispebjerg and Frederiksberg Hospital, Copenhagen, Denmark; Lauren Lissner: University of Gothenburg, Gothenburg, Sweden; Mieczyslaw Litwin: Children's Memorial Health Institute, Warsaw, Poland; Lijuan Liu: Capital Medical University Beijing Tongren Hospital, Beijing, China; Wei-Cheng Lo: Taipei Medical University, Taipei, Taiwan; Helle-Mai Loit: National Institute for Health Development, Tallinn, Estonia; Khuong Quynh Long: Hanoi University of Public Health, Hanoi, Viet Nam; Luis Lopes: University of Porto, Porto, Portugal; Oscar Lopes: Sports Medical Center of Minho, Braga, Portugal; Esther Lopez-Garcia: Universidad Autónoma de Madrid CIBERESP, Madrid, Spain; Tania Lopez: Universidad San Martín de Porres, Lima, Peru; Paulo A Lotufo: University of São Paulo, São Paulo, Brazil; José Eugenio Lozano: Consejería de Sanidad Junta de Castilla y León, Valladolid, Spain; Janice L Lukrafka: Universidade Federal de Ciências da Saúde de Porto Alegre, Porto Alegre, Brazil; Dalia Luksiene: Lithuanian University of Health Sciences, Kaunas, Lithuania; Annamari Lundqvist: Finnish Institute for Health and Welfare, Helsinki, Finland; Robert Lundqvist: Norrbotten County Council, Luleå, Sweden; Nuno Lunet: University of Porto, Porto, Portugal; Charles Lunogelo: Ilembula Lutheran Hospital, Ilembula, United Republic of Tanzania; Michala Lustigová: Charles University, Prague, Czech Republic; National Institute of Public Health, Prague, Czech Republic; Edyta Łuszczki: University of Rzeszów, Rzeszów, Poland; Guansheng Ma: Peking University, Beijing, China; Jun Ma: Peking University, Beijing, China; Xu Ma: National Research Institute for Health and Family Planning, Beijing, China; George LL Machado-Coelho: Universidade Federal de Ouro Preto, Ouro Preto, Brazil; Aristides M Machado-Rodrigues: University of Coimbra, Coimbra, Portugal; Luisa M Macieira: Coimbra University Hospital Center, Coimbra, Portugal; Ahmed A Madar: University of Oslo, Oslo, Norway; Stefania Maggi: Institute of Neuroscience of the National Research Council, Padua, Italy; Dianna J Magliano: Baker Heart and Diabetes Institute, Melbourne, Australia; Sara Magnacca: Mediterranea Cardiocentro, Naples, Italy; Emmanuella Magriplis: Agricultural University of Athens, Athens, Greece; Gowri Mahasampath: Christian Medical College, Vellore, India; Bernard Maire: French National Research Institute for Sustainable Development, Montpellier, France; Marjeta Majer: University of Zagreb, Zagreb, Croatia; Marcia Makdisse: Hospital Israelita Albert Einstein, São Paulo, Brazil; Päivi Mäki: Finnish Institute for Health and Welfare, Helsinki, Finland; Fatemeh Malekzadeh: Tehran University of Medical Sciences, Tehran, Islamic Republic of Iran; Reza Malekzadeh: Tehran University of Medical Sciences, Tehran, Islamic Republic of Iran; Rahul Malhotra: Duke-NUS Medical School, Singapore, Singapore; Kodavanti Mallikharjuna Rao: ICMR - National Institute of Nutrition, Hyderabad, India; Sofia K Malyutina: SB RAS Federal Research Center Institute of Cytology and Genetics, Novosibirsk, Russian Federation; Lynell V Maniego: Food and Nutrition Research Institute, Taguig, Philippines; Yannis Manios: Harokopio University, Athens, Greece; Jim I Mann: University of Otago, Dunedin, New Zealand; Fariborz Mansour-Ghanaei: Guilan University of Medical Sciences, Rasht, Islamic Republic of Iran; Enzo Manzato: University of Padua, Padua, Italy; Paula Margozzini: Pontificia Universidad Católica de Chile, Santiago, Chile; Anastasia Markaki: Hellenic Mediterranean University, Siteia, Greece; Oonagh Markey: Loughborough University, Loughborough, United Kingdom; Eliza Markidou Ioannidou: Ministry of Health, Nicosia, Cyprus; Pedro Marques-Vidal: Lausanne University Hospital, Lausanne, Switzerland; Larissa Pruner Marques: Escola Nacional de Saúde Pública Sergio Arouca, Rio de Janeiro, Brazil; Jaume Marrugat: CIBERCV, Madrid, Spain; Institut Hospital del Mar d'Investigacions Mèdiques, Barcelona, Spain; Yves Martin-Prevel: French National Research

Institute for Sustainable Development, Montpellier, France; Rosemarie Martin: Mary Immaculate College, Limerick, Ireland; Reynaldo Martorell: Emory University, Atlanta, United States; Eva Martos: Hungarian Society of Sports Medicine, Budapest, Hungary; Katharina Maruszczak: Paracelsus Medical University, Salzburg, Austria; Stefano Marventano: University of Catania, Catania, Italy; Luis P Mascarenhas: Universidade Estadual do Centro-Oeste, Guarapuava, Brazil; Shariq R Masoodi: Sher-i-Kashmir Institute of Medical Sciences, Srinagar, India; Ellisiv B Mathiesen: UiT The Arctic University of Norway, Tromsø, Norway; Prashant Mathur: ICMR - National Centre for Disease Informatics and Research, Bengaluru, India; Alicia Matijasevich: University of São Paulo, São Paulo, Brazil; Tandi E Matsha: Cape Peninsula University of Technology, Cape Town, South Africa; Christina Mavrogianni: Harokopio University, Athens, Greece; Artur Mazur: University of Rzeszów, Rzeszów, Poland; Jean Claude N Mbanya: University of Yaoundé 1, Yaoundé, Cameroon; Shelly R McFarlane: The University of the West Indies, Kingston, Jamaica; Stephen T McGarvey: Brown University, Providence, United States; Martin McKee: London School of Hygiene & Tropical Medicine, London, United Kingdom; Stela McLachlan: University of Edinburgh, Edinburgh, United Kingdom; Rachael M McLean: University of Otago, Dunedin, New Zealand; Scott B McLean: Statistics Canada, Ottawa, Canada; Breige A McNulty: University College Dublin, Dublin, Ireland; Sounnia Mediene Benchekor: University Oran 1, Oran, Algeria; Jurate Medzioniene: Lithuanian University of Health Sciences, Kaunas, Lithuania; Parinaz Mehdipour: Non-Communicable Diseases Research Center, Tehran, Islamic Republic of Iran; Kirsten Mehlig: University of Gothenburg, Gothenburg, Sweden; Amir Houshang Mehrparvar: Shahid Sadoughi University of Medical Sciences, Tehran, Islamic Republic of Iran; Aline Meirhaeghe: Institut National de la Santé et de la Recherche Médicale, Lille, France; Jørgen Meisfjord: Norwegian Institute of Public Health, Oslo, Norway; Christa Meisinger: Helmholtz Zentrum München, Munich, Germany; Ana Maria B Menezes: Federal University of Pelotas, Pelotas, Brazil; Geetha R Menon: ICMR - National Institute of Medical Statistics, New Delhi, India; Gert BM Mensink: Robert Koch Institute, Berlin, Germany; Maria Teresa Menzano: Ministero della Salute DG Prevenzione Sanitaria, Rome, Italy; Alibek Mereke: Al-Farabi Kazakh National University, Almaty, Kazakhstan; Indrapal I Meshram: ICMR - National Institute of Nutrition, Hyderabad, India; Andres Metspalu: University of Tartu, Tartu, Estonia; Haakon E Meyer: University of Oslo, Oslo, Norway; Jie Mi: Capital Institute of Pediatrics, Beijing, China; Kim F Michaelsen: University of Copenhagen, Copenhagen, Denmark; Nathalie Michels: Ghent University, Ghent, Belgium; Kairit Mikkel: University of Tartu, Tartu, Estonia; Karolina Milkowska: Jagiellonian University Medical College, Kraków, Poland; Jody C Miller: University of Otago, Dunedin, New Zealand; Cláudia S Minderico: Universidade de Lisboa, Lisbon, Portugal; GK Mini: Women's Social and Health Studies Foundation, Trivandrum, India; Juan Francisco Miquel: Pontificia Universidad Católica de Chile, Santiago, Chile; Mohammad Reza Mirjalili: Shahid Sadoughi University of Medical Sciences, Tehran, Islamic Republic of Iran; Daphne Mirkopoulou: Democritus University, Alexandroupolis, Greece; Erkin Mirrakhimov: Kyrgyz State Medical Academy, Bishkek, Kyrgyzstan; Marjeta Mišigoj-Durakovic: University of Zagreb, Zagreb, Croatia; Antonio Mistretta: University of Catania, Catania, Italy; Veronica Mocanu: Grigore T Popa University of Medicine and Pharmacy, Iasi, Romania; Pietro A Modesti: Università degli Studi di Firenze, Florence, Italy; Sahar Saeedi Moghaddam: Non-Communicable Diseases Research Center, Tehran, Islamic Republic of Iran; Bahram Mohajer: Non-Communicable Diseases Research Center, Tehran, Islamic Republic of Iran; Mostafa K Mohamed: Ain Shams University, Cairo, Egypt; Shukri F Mohamed: African Population and Health Research Center, Nairobi, Kenya; Kazem Mohammad: Tehran University of Medical Sciences, Tehran, Islamic Republic of Iran; Zahra Mohammadi: Tehran University of Medical Sciences, Tehran, Islamic Republic of Iran; Noushin Mohammadifard: Isfahan Cardiovascular Research Center, Isfahan, Islamic Republic of Iran; Reza Mohammadpourhodki: Mashhad University of Medical Sciences, Mashhad, Islamic Republic of Iran; Viswanathan Mohan: Madras Diabetes Research Foundation, Chennai, India; Salim Mohanna: Universidad Peruana Cayetano Heredia, Lima, Peru; Muhammad Fadhli Mohd Yusoff: Ministry of Health, Kuala Lumpur, Malaysia; Iraj Mohebbi: Urmia University of Medical Sciences, Urmia, Islamic Republic of Iran; Farnam Mohebi: Non-Communicable Diseases Research Center, Tehran, Islamic Republic of Iran; Marie Moitry: University of Strasbourg, Strasbourg, France; Strasbourg University Hospital, Strasbourg, France; Drude Molbo: University of Copenhagen, Copenhagen, Denmark; Line T Møllehave: Bispebjerg and Frederiksberg Hospital,

Copenhagen, Denmark; Niels C Møller: University of Southern Denmark, Odense, Denmark; Dénes Molnár: University of Pécs, Pécs, Hungary; Amirabbas Momenan: Shahid Beheshti University of Medical Sciences, Tehran, Islamic Republic of Iran; Charles K Mondo: Mulago Hospital, Kampala, Uganda; Michele Monroy-Valle: University of San Carlos of Guatemala, Guatemala City, Guatemala; Eric Monterrubio-Flores: National Institute of Public Health, Cuernavaca, Mexico; Kotsedi Daniel K Monyeki: University of Limpopo, Sovenga, South Africa; Jin Soo Moon: Seoul National University, Seoul, Republic of Korea; Mahmood Moosazadeh: Mazandaran University of Medical Sciences, Sari, Islamic Republic of Iran; Leila B Moreira: Universidade Federal do Rio Grande do Sul, Porto Alegre, Brazil; Alain Morejon: University of Medical Sciences of Cienfuegos, Cienfuegos, Cuba; Luis A Moreno: University of Zaragoza, Zaragoza, Spain; CIBEROBN, Zaragoza, Spain; Karen Morgan: Royal College of Surgeons in Ireland, Dublin, Ireland; Suzanne N Morin: McGill University, Montreal, Canada; Erik Lykke Mortensen: University of Copenhagen, Copenhagen, Denmark; George Moschonis: La Trobe University, Melbourne, Australia; Malgorzata Mossakowska: International Institute of Molecular and Cell Biology, Warsaw, Poland; Aya Mostafa: Ain Shams University, Cairo, Egypt; Anabela Mota-Pinto: University of Coimbra, Coimbra, Portugal; Jorge Mota: University of Porto, Porto, Portugal; Mohammad Esmaeel Motlagh: Ahvaz Jundishapur University of Medical Sciences, Ahvaz, Islamic Republic of Iran; Jorge Motta: Instituto Conmemorativo Gorgas de Estudios de la Salud, Panama City, Panama; Marcos André Moura-dos-Santos: University of Pernambuco, Recife, Brazil; Malay K Mridha: BRAC James P Grant School of Public Health, Dhaka, Bangladesh; Kelias P Msyamboza: World Health Organization Country Office, Lilongwe, Malawi; Thet Thet Mu: Department of Public Health, Nay Pyi Taw, Myanmar; Magdalena Muc: University of Coimbra, Coimbra, Portugal; Boban Mugoša: Institute of Public Health, Podgorica, Montenegro; Maria L Muiesan: University of Brescia, Brescia, Italy; Parvina Mukhtorova: Ministry of Health and Social Protection, Dushanbe, Tajikistan; Martina Müller-Nurasyid: Helmholtz Zentrum München, Munich, Germany; Neil Murphy: International Agency for Research on Cancer, Lyon, France; Jaakko Mursu: University of Eastern Finland, Kuopio, Finland; Elaine M Murtagh: University of Limerick, Limerick, Ireland; Kamarul Imran Musa: Universiti Sains Malaysia, Kelantan, Malaysia; Sanja Music Milanovic: Croatian Institute of Public Health, Zagreb, Croatia; University of Zagreb, Zagreb, Croatia; Vera Musil: University of Zagreb, Zagreb, Croatia; Norlaila Mustafa: Universiti Kebangsaan Malaysia, Kuala Lumpur, Malaysia; Iraj Nabipour: Bushehr University of Medical Sciences, Bushehr, Islamic Republic of Iran; Shohreh Naderimagham: Non-Communicable Diseases Research Center, Tehran, Islamic Republic of Iran; Gabriele Nagel: Ulm University, Ulm, Germany; Balkish M Naidu: Department of Statistics, Kuala Lumpur, Malaysia; Farid Najafi: Kermanshah University of Medical Sciences, Kermanshah, Islamic Republic of Iran; Harunobu Nakamura: Kobe University, Kobe, Japan; Jana Námešná: Banska Bystrica Regional Authority of Public Health, Banska Bystrica, Slovakia; Ei Ei K Nang: National University of Singapore, Singapore, Singapore; Vinay B Nangia: Suraj Eye Institute, Nagpur, India; Martin Nankap: UNICEF, Yaoundé, Cameroon; Sameer Narake: Healis-Sekhsaria Institute for Public Health, Navi Mumbai, India; Paola Nardone: Istituto Superiore di Sanità, Rome, Italy; Matthias Nauck: University Medicine Greifswald, Greifswald, Germany; William A Neal: West Virginia University, Morgantown, United States; Azim Nejatizadeh: Hormozgan University of Medical Sciences, Bandar Abbas, Islamic Republic of Iran; Chandini Nekkantti: University of New South Wales, Sydney, Australia; Keiu Nelis: National Institute for Health Development, Tallinn, Estonia; Liis Nelis: National Institute for Health Development, Tallinn, Estonia; Ilona Nenko: Jagiellonian University Medical College, Kraków, Poland; Martin Neovius: Karolinska Institutet, Stockholm, Sweden; Flavio Nervi: Pontificia Universidad Católica de Chile, Santiago, Chile; Chung T Nguyen: National Institute of Hygiene and Epidemiology, Hanoi, Viet Nam; Nguyen D Nguyen: University of Medicine and Pharmacy at Ho Chi Minh City, Ho Chi Minh City, Viet Nam; Quang Ngoc Nguyen: Hanoi Medical University, Hanoi, Viet Nam; Ramfis E Nieto-Martínez: LifeDoc Health, Memphis, United States; Yury P Nikitin: SB RAS Federal Research Center Institute of Cytology and Genetics, Novosibirsk, Russian Federation; Guang Ning: Shanghai Jiao-Tong University School of Medicine, Shanghai, China; Toshiharu Ninomiya: Kyushu University, Fukuoka, Japan; Sania Nishtar: Heartfile, Islamabad, Pakistan; Marianna Noale: Institute of Neuroscience of the National Research Council, Padua, Italy; Oscar A Noboa: Universidad de la República, Montevideo, Uruguay; Helena Nogueira: University of Coimbra, Coimbra, Portugal; Teresa

Norat: Imperial College London, London, United Kingdom; Maria Nordendahl: Umeå University, Umeå, Sweden; Børge G Nordestgaard: Copenhagen University Hospital, Copenhagen, Denmark; University of Copenhagen, Copenhagen, Denmark; Davide Noto: University of Palermo, Palermo, Italy; Natalia Nowak-Szczepanska: PASs Hirszfeld Institute of Immunology and Experimental Therapy, Wroclaw, Poland; Mohannad Al Nsour: Eastern Mediterranean Public Health Network, Amman, Jordan; Irfan Nuhoglu: Karadeniz Technical University, Trabzon, Turkey; Eha Nurk: National Institute for Health Development, Tallinn, Estonia; Terence W O'Neill: University of Manchester, Manchester, United Kingdom; Dermot O'Reilly: Queen's University of Belfast, Belfast, United Kingdom; Galina Obreja: State University of Medicine and Pharmacy, Chisinau, Moldova; Caleb Ochimana: Harvard TH Chan School of Public Health, Boston, United States; Angélica M Ochoa-Avilés: Universidad de Cuenca, Cuenca, Ecuador; Eiji Oda: Tachikawa General Hospital, Nagaoka, Japan; Kyungwon Oh: Korea Centers for Disease Control and Prevention, Cheongju-si, Republic of Korea; Kumiko Ohara: Kindai University, Osaka-Sayama, Japan; Claes Ohlsson: University of Gothenburg, Gothenburg, Sweden; Sahlgrenska University Hospital, Gothenburg, Sweden; Ryutaro Ohtsuka: Japan Wildlife Research Center, Tokyo, Japan; Örn Olafsson: Icelandic Heart Association, Kopavogur, Iceland; Maria Teresa A Olinto: Universidade do Vale do Rio dos Sinos, São Leopoldo, Brazil; Isabel O Oliveira: Federal University of Pelotas, Pelotas, Brazil; Mohd Azahadi Omar: Ministry of Health, Kuala Lumpur, Malaysia; Altan Onat: Istanbul University, Istanbul, Turkey; Sok King Ong: Ministry of Health, Bandar Seri Begawan, Brunei Darussalam; Lariane M Ono: Universidade Federal do Paraná, Curitiba, Brazil; Pedro Ordunez: Pan American Health Organization, Washington DC, United States; Rui Ornelas: University of Madeira, Funchal, Portugal; Ana P Ortiz: University of Puerto Rico, San Juan, Puerto Rico; Pedro J Ortiz: Universidad Peruana Cayetano Heredia, Lima, Peru; Merete Osler: Center for Clinical Research and Prevention, Glostrup, Denmark; Clive Osmond: MRC Lifecourse Epidemiology Unit, Southampton, United Kingdom; Sergej M Ostojic: University of Novi Sad, Novi Sad, Serbia; Afshin Ostovar: Tehran University of Medical Sciences, Tehran, Islamic Republic of Iran; Johanna A Otero: Fundación Oftalmológica de Santander, Bucaramanga, Colombia; Kim Overvad: Aarhus University, Aarhus, Denmark; Ellis Owusu-Dabo: Kwame Nkrumah University of Science and Technology, Kumasi, Ghana; Fred Michel Paccaud: UniSanté, Lausanne, Switzerland; Cristina Padez: University of Coimbra, Coimbra, Portugal; Ioannis Pagkalos: International Hellenic University, Thessaloniki, Greece; Elena Pahomova: University of Latvia, Riga, Latvia; Karina Mary de Paiva: Federal University of Santa Catarina, Florianópolis, Brazil; Andrzej Pajak: Jagiellonian University Medical College, Kraków, Poland; Domenico Palli: Institute for Cancer Research, Prevention and Clinical Network, Florence, Italy; Alberto Palloni: University of Wisconsin-Madison, Madison, United States; Luigi Palmieri: Istituto Superiore di Sanità, Rome, Italy; Wen-Harn Pan: Academia Sinica, Taipei, Taiwan; Songhomitra Panda-Jonas: Ruprecht-Karls-University of Heidelberg, Heidelberg, Germany; Arvind Pandey: ICMR - National Institute of Medical Statistics, New Delhi, India; Francesco Panza: IRCCS Ente Ospedaliero Specializzato in Gastroenterologia S. de Bellis, Bari, Italy; Dimitrios Papandreou: Zayed University, Abu Dhabi, United Arab Emirates; Soon-Woo Park: Catholic University of Daegu, Daegu, Republic of Korea; Suyeon Park: Korea Centers for Disease Control and Prevention, Cheongju-si, Republic of Korea; Winsome R Parnell: University of Otago, Dunedin, New Zealand; Mahboubeh Parsaeian: Tehran University of Medical Sciences, Tehran, Islamic Republic of Iran; Ionela M Pascanu: University of Medicine, Pharmacy, Science and Technology of Târgu Mures, Târgu Mures, Romania; Patrick Pasquet: UMR CNRS-MNHN 7206 Eco-anthropologie, Paris, France; Nikhil D Patel: Jivandeep Hospital, Anand, India; Ivan Pecin: University Hospital Center Zagreb, Zagreb, Croatia; Mangesh S Pednekar: Healis-Sekhsaria Institute for Public Health, Navi Mumbai, India; Nasheeta Peer: South African Medical Research Council, Durban, South Africa; Gao Pei: Peking University, Beijing, China; Sergio Viana Peixoto: Oswaldo Cruz Foundation Rene Rachou Research Institute, Belo Horizonte, Brazil; Markku Peltonen: Finnish Institute for Health and Welfare, Helsinki, Finland; Alexandre C Pereira: University of São Paulo, São Paulo, Brazil; Marco A Peres: National Dental Care Centre Singapore, Singapore, Singapore; Napoléon Pérez-Farinós: Universidad de Málaga, Malaga, Spain; Cynthia M Pérez: University of Puerto Rico, San Juan, Puerto Rico; Valentina Peterkova: Endocrinology Research Centre, Moscow, Russian Federation; Annette Peters: Helmholtz Zentrum München, Munich, Germany; Astrid Petersmann: University Medicine Greifswald,

Greifswald, Germany; Janina Petkeviciene: Lithuanian University of Health Sciences, Kaunas, Lithuania; Ausra Petrauskiene: Lithuanian University of Health Sciences, Kaunas, Lithuania; Emanuela Pettenuzzo: University Hospital of Varese, Varese, Italy; Niloofar Peykari: Ministry of Health and Medical Education, Tehran, Islamic Republic of Iran; Son Thai Pham: Vietnam National Heart Institute, Hanoi, Viet Nam; Rafael N Pichardo: Clínica de Medicina Avanzada Dr. Abel González, Santo Domingo, Dominican Republic; Daniela Pierannunzio: Istituto Superiore di Sanità, Rome, Italy; Iris Pigeot: Leibniz Institute for Prevention Research and Epidemiology - BIPS, Bremen, Germany; Hynek Pikhart: University College London, London, United Kingdom; Aida Pilav: University of Sarajevo, Sarajevo, Bosnia and Herzegovina; Lorenza Pilotto: Cardiovascular Prevention Centre Udine, Udine, Italy; Francesco Pistelli: Pisa University Hospital, Pisa, Italy; Freda Pitakaka: Ministry of Health and Medical Services, Honiara, Solomon Islands; Aleksandra Piwonska: National Institute of Cardiology, Warsaw, Poland; Andreia N Pizarro: University of Porto, Porto, Portugal; Pedro Plans-Rubió: Public Health Agency of Catalonia, Barcelona, Spain; Bee Koon Poh: Universiti Kebangsaan Malaysia, Kuala Lumpur, Malaysia; Hermann Pohlabeln: Leibniz Institute for Prevention Research and Epidemiology - BIPS, Bremen, Germany; Raluca M Pop: University of Medicine, Pharmacy, Science and Technology of Târgu Mures, Târgu Mures, Romania; Stevo R Popovic: University of Montenegro, Niksic, Montenegro; Miquel Porta: Institut Hospital del Mar d'Investigacions Mèdiques, Barcelona, Spain; Georg Posch: Agency for Preventive and Social Medicine, Bregenz, Austria; Anil Poudyal: Nepal Health Research Council, Kathmandu, Nepal; Dimitrios Poulimeneas: International Hellenic University, Thessaloniki, Greece; Hamed Pouraram: Tehran University of Medical Sciences, Tehran, Islamic Republic of Iran; Farhad Pourfarzi: Ardabil University of Medical Sciences, Ardabil, Islamic Republic of Iran; Akram Pourshams: Tehran University of Medical Sciences, Tehran, Islamic Republic of Iran; Hossein Poustchi: Tehran University of Medical Sciences, Tehran, Islamic Republic of Iran; Rajendra Pradeepa: Madras Diabetes Research Foundation, Chennai, India; Alison J Price: London School of Hygiene & Tropical Medicine, London, United Kingdom; Jacqueline F Price: University of Edinburgh, Edinburgh, United Kingdom; Rui Providencia: University College London, London, United Kingdom; Jardena J Puder: Lausanne University Hospital, Lausanne, Switzerland; Iveta Pudule: Centre for Disease Prevention and Control, Riga, Latvia; Soile E Puhakka: University of Oulu, Oulu, Finland; Oulu Deaconess Institute Foundation, Oulu, Finland; Maria Puiu: Victor Babes University of Medicine and Pharmacy Timisoara, Timisoara, Romania; Margus Punab: Tartu University Clinics, Tartu, Estonia; Radwan F Qasrawi: Al-Quds University, East Jerusalem, State of Palestine; Mostafa Qorbani: Alborz University of Medical Sciences, Karaj, Islamic Republic of Iran; Tran Quoc Bao: Ministry of Health, Hanoi, Viet Nam; Ivana Radic: University of Novi Sad, Novi Sad, Serbia; Ricardas Radisauskas: Lithuanian University of Health Sciences, Kaunas, Lithuania; Salar Rahimikazerooni: Shiraz University of Medical Sciences, Shiraz, Islamic Republic of Iran; Mahfuzar Rahman: Pure Earth, Dhaka, Bangladesh; Mahmudur Rahman: Institute of Epidemiology Disease Control and Research, Dhaka, Bangladesh; Olli Raitakari: University of Turku, Turku, Finland; Manu Raj: Amrita Institute of Medical Sciences, Cochin, India; Ellina Rakhimova: Ufa Eye Research Institute, Ufa, Russian Federation; Sherali Rakhmatulloev: Ministry of Health and Social Protection, Dushanbe, Tajikistan; Ivo Rakovac: World Health Organization Regional Office for Europe, Moscow, Russian Federation; Sudha Ramachandra Rao: National Institute of Epidemiology, Chennai, India; Ambady Ramachandran: India Diabetes Research Foundation, Chennai, India; Jacqueline Ramke: University of Auckland, Auckland, New Zealand; Elisabete Ramos: University of Porto Medical School, Porto, Portugal; Rafel Ramos: Institut Universitari d'Investigació en Atenció Primària Jordi Gol, Girona, Spain; Lekhraj Rampal: Universiti Putra Malaysia, Serdang, Malaysia; Sanjay Rampal: University of Malaya, Kuala Lumpur, Malaysia; Vayia Rarra: Sotiria Hospital, Athens, Greece; Ramon A Rascon-Pacheco: Instituto Mexicano del Seguro Social, Mexico City, Mexico; Mette Rasmussen: University of Southern Denmark, Odense, Denmark; Cassiano Ricardo Rech: Federal University of Santa Catarina, Florianópolis, Brazil; Josep Redon: University of Valencia, Valencia, Spain; Paul Ferdinand M Reganit: University of the Philippines, Manila, Philippines; Valéria Regecová: Slovak Academy of Sciences, Bratislava, Slovakia; Luis Revilla: Universidad San Martín de Porres, Lima, Peru; Abbas Rezaianzadeh: Shiraz University of Medical Sciences, Shiraz, Islamic Republic of Iran; Lourdes Ribas-Barba: Nutrition Research Foundation, Barcelona, Spain; Robespierre Ribeiro: Minas Gerais State Secretariat for Health, Belo Horizonte, Brazil; Elio

Riboli: Imperial College London, London, United Kingdom; Adrian Richter: University Medicine Greifswald, Greifswald, Germany; Fernando Rigo: CS S. Agustín Ibsalut, Palma, Spain; Natascia Rinaldo: University of Ferrara, Ferrara, Italy; Tobias F Rinke de Wit: Amsterdam Institute for Global Health and Development, Amsterdam, Netherlands; Ana Rito: National Institute of Health Doutor Ricardo Jorge, Lisbon, Portugal; Raphael M Ritti-Dias: Universidade Nove de Julho, São Paulo, Brazil; Juan A Rivera: National Institute of Public Health, Cuernavaca, Mexico; Cynthia Robitaille: Public Health Agency of Canada, Ottawa, Canada; Romana Roccaldo: Council for Agricultural Research and Economics, Rome, Italy; Daniela Rodrigues: University of Coimbra, Coimbra, Portugal; Fernando Rodríguez-Artalejo: Universidad Autónoma de Madrid CIBERESP, Madrid, Spain; María del Cristo Rodriguez-Perez: Canarian Health Service, Tenerife, Spain; Laura A Rodríguez-Villamizar: Universidad Industrial de Santander, Bucaramanga, Colombia; Ulla Roggenbuck: University of Duisburg-Essen, Duisburg, Germany; Rosalba Rojas-Martinez: National Institute of Public Health, Cuernavaca, Mexico; Nipa Rojroongwasinkul: Mahidol University, Nakhon Pathom, Thailand; Dora Romaguera: CIBEROBN, Madrid, Spain; Elisabetta L Romeo: Associazione Calabrese di Epatologia, Reggio Calabria, Italy; Rafaela V Rosario: University of Minho, Braga, Portugal; Annika Rosengren: University of Gothenburg, Gothenburg, Sweden; Sahlgrenska University Hospital, Gothenburg, Sweden; Ian Rouse: Fiji National University, Suva, Fiji; Joel GR Roy: Statistics Canada, Ottawa, Canada; Adolfo Rubinstein: Institute for Clinical Effectiveness and Health Policy, Buenos Aires, Argentina; Frank J Rühli: University of Zurich, Zurich, Switzerland; Jean-Bernard Ruidavets: Toulouse University School of Medicine, Toulouse, France; Blanca Sandra Ruiz-Betancourt: Instituto Mexicano del Seguro Social, Mexico City, Mexico; Maria Ruiz-Castell: Luxembourg Institute of Health, Strassen, Luxembourg; Emma Ruiz Moreno: Spanish Nutrition Foundation, Madrid, Spain; Iuliia A Rusakova: Ufa Eye Research Institute, Ufa, Russian Federation; Kenisha Russell Jonsson: Public Health Agency of Sweden, Stockholm, Sweden; Paola Russo: Institute of Food Sciences of the National Research Council, Avellino, Italy; Petra Rust: University of Vienna, Vienna, Austria; Marcin Rutkowski: Medical University of Gdansk, Gdansk, Poland; Charumathi Sabanayagam: Singapore Eye Research Institute, Singapore, Singapore; Elena Sacchini: Authority Sanitaria San Marino, San Marino, San Marino; Harshpal S Sachdev: Sitaram Bhartia Institute of Science and Research, New Delhi, India; Alireza Sadjadi: Tehran University of Medical Sciences, Tehran, Islamic Republic of Iran; Ali Reza Safarpour: Shiraz University of Medical Sciences, Shiraz, Islamic Republic of Iran; Saeid Safiri: Tabriz University of Medical Sciences, Tabriz, Islamic Republic of Iran; Nader Saki: Ahvaz Jundishapur University of Medical Sciences, Ahvaz, Islamic Republic of Iran; Benoit Salanave: French Public Health Agency, St Maurice, France; Eduardo Salazar Martinez: National Institute of Public Health, Cuernavaca, Mexico; Diego Salmerón: CIBER en Epidemiología y Salud Pública, Murcia, Spain; Veikko Salomaa: Finnish Institute for Health and Welfare, Helsinki, Finland; Jukka T Salonen: University of Helsinki, Helsinki, Finland; Massimo Salvetti: University of Brescia, Brescia, Italy; Margarita Samoutian: Kindergarten of Avlonari, Evia, Greece; Jose Sánchez-Abanto: National Institute of Health, Lima, Peru; : Ministry of Health, Jakarta, Indonesia; Susana Sans: Catalan Department of Health, Barcelona, Spain; Loreto Santa Marina: Biodonostia Health Research Institute, San Sebastian, Spain; Diana A Santos: Universidade de Lisboa, Lisbon, Portugal; Ina S Santos: Federal University of Pelotas, Pelotas, Brazil; Lèlita C Santos: Coimbra University Hospital Center, Coimbra, Portugal; Maria Paula Santos: University of Porto, Porto, Portugal; Osvaldo Santos: Instituto de Saúde Ambiental, Lisbon, Portugal; Rute Santos: University of Porto, Porto, Portugal; Sara Santos Sanz: Spanish Agency for Food Safety and Nutrition, Madrid, Spain; Jouko L Saramies: South Karelia Social and Health Care District, Lappeenranta, Finland; Luis B Sardinha: Universidade de Lisboa, Lisbon, Portugal; Nizal Sarrafzadegan: Isfahan Cardiovascular Research Center, Isfahan, Islamic Republic of Iran; Thirunavukkarasu Sathish: McMaster University, Hamilton, Canada; Kai-Uwe Saum: German Cancer Research Center, Heidelberg, Germany; Savvas Savva: Research and Education Institute of Child Health, Nicosia, Cyprus; Mathilde Savy: French National Research Institute for Sustainable Development, Montpellier, France; Norie Sawada: National Cancer Center, Tokyo, Japan; Mariana Sbaraini: Universidade Federal do Rio Grande do Sul, Porto Alegre, Brazil; Marcia Scazufca: University of São Paulo Clinics Hospital, São Paulo, Brazil; Beatriz D Schaan: Universidade Federal do Rio Grande do Sul, Porto Alegre, Brazil; Angelika Schaffrath Rosario: Robert Koch Institute, Berlin, Germany; Herman Schargrodsky: Hospital Italiano de Buenos

Aires, Buenos Aires, Argentina; Anja Schienkiewitz: Robert Koch Institute, Berlin, Germany; Sabine Schipf: University Medicine Greifswald, Greifswald, Germany; Carsten O Schmidt: University Medicine Greifswald, Greifswald, Germany; Ida Maria Schmidt: Rigshospitalet, Copenhagen, Denmark; Peter Schnohr: Copenhagen University Hospital, Copenhagen, Denmark; Ben Schöttker: German Cancer Research Center, Heidelberg, Germany; Sara Schramm: University of Duisburg-Essen, Duisburg, Germany; Stine Schramm: University of Southern Denmark, Odense, Denmark; Helmut Schröder: CIBER en Epidemiología y Salud Pública, Barcelona, Spain; Constance Schultsz: Academic Medical Center of University of Amsterdam, Amsterdam, Netherlands; Aletta E Schutte: University of New South Wales, Sydney, Australia; The George Institute for Global Health, Sydney, Australia; Aye Aye Sein: Ministry of Health and Sports, Nay Pyi Taw, Myanmar; Rusidah Selamat: Ministry of Health, Kuala Lumpur, Malaysia; Vedrana Sember: University of Ljubljana, Ljubljana, Slovenia; Abhijit Sen: Center for Oral Health Services and Research Mid-Norway, Trondheim, Norway; Idowu O Senbanjo: Lagos State University College of Medicine, Lagos, Nigeria; Sadaf G Sepanlou: Tehran University of Medical Sciences, Tehran, Islamic Republic of Iran; Victor Sequera: Ministry of Public Health, Asunción, Paraguay; Luis Serra-Majem: University of Las Palmas de Gran Canaria, Las Palmas de Gran Canaria, Spain; Jennifer Servais: Statistics Canada, Ottawa, Canada; Ludmila Ševcíková: Comenius University, Bratislava, Slovakia; Svetlana A Shalnova: National Research Centre for Preventive Medicine, Moscow, Russian Federation; Teresa Shamah-Levy: National Institute of Public Health, Cuernavaca, Mexico; Morteza Shamshirgaran: Neyshabur University of Medical Sciences, Neyshabur, Islamic Republic of Iran; Coimbatore Subramaniam Shanthirani: Madras Diabetes Research Foundation, Chennai, India; Maryam Sharafkhah: Tehran University of Medical Sciences, Tehran, Islamic Republic of Iran; Sanjib K Sharma: B P Koirala Institute of Health Sciences, Dharan, Nepal; Jonathan E Shaw: Baker Heart and Diabetes Institute, Melbourne, Australia; Amaneh Shayanrad: Tehran University of Medical Sciences, Tehran, Islamic Republic of Iran; Ali Akbar Shayesteh: Ahvaz Jundishapur University of Medical Sciences, Ahvaz, Islamic Republic of Iran; Lela Shengelia: National Center for Disease Control and Public Health, Tbilisi, Georgia; Zumin Shi: Qatar University, Doha, Qatar; Kenji Shibuya: King's College London, London, United Kingdom; Hana Shimizu-Furusawa: Nippon Medical School, Tokyo, Japan; Dong Wook Shin: Sungkyunkwan University, Seoul, Republic of Korea; Majid Shirani: Shahrekord University of Medical Sciences, Shahrekord, Islamic Republic of Iran; Rahman Shiri: Finnish Institute of Occupational Health, Helsinki, Finland; Namuna Shrestha: Public Health Promotion and Development Organization, Kathmandu, Nepal; Khairil Si-Ramlee: Ministry of Health, Bandar Seri Begawan, Brunei Darussalam; Alfonso Siani: Institute of Food Sciences of the National Research Council, Avellino, Italy; Rosalynn Siantar: Singapore Eye Research Institute, Singapore, Singapore; Abla M Sibai: American University of Beirut, Beirut, Lebanon; Antonio M Silva: Federal University of Maranhão, São Luís, Brazil; Diego Augusto Santos Silva: Federal University of Santa Catarina, Florianópolis, Brazil; Mary Simon: India Diabetes Research Foundation, Chennai, India; Judith Simons: St Vincent's Hospital, Sydney, Australia; Leon A Simons: University of New South Wales, Sydney, Australia; Agneta Sjöberg: University of Gothenburg, Gothenburg, Sweden; Michael Sjöström: Karolinska Institutet, Stockholm, Sweden; Gry Skodje: Nes Municipality, Aarnes, Norway; Jolanta Slowikowska-Hilczer: Medical University of Lodz, Lodz, Poland; Przemyslaw Slusarczyk: International Institute of Molecular and Cell Biology, Warsaw, Poland; Liam Smeeth: London School of Hygiene & Tropical Medicine, London, United Kingdom; Hung-Kwan So: University of Hong Kong, Hong Kong, China; Fernanda Cunha Soares: University of Pernambuco, Recife, Brazil; Grzegorz Sobek: University of Rzeszów, Rzeszów, Poland; Eugène Sobngwi: University of Yaoundé 1, Yaoundé, Cameroon; Morten Sodemann: University of Southern Denmark, Odense, Denmark; Stefan Söderberg: Umeå University, Umeå, Sweden; Moesijanti YE Soekatri: Health Polytechnic Jakarta II Institute, Jakarta, Indonesia; Agustinus Soemantri: Diponegoro University, Semarang, Indonesia; Reecha Sofat: University College London, London, United Kingdom; Vincenzo Solfrizzi: University of Bari, Bari, Italy; Mohammad Hossein Somi: Tabriz University of Medical Sciences, Tabriz, Islamic Republic of Iran; Emily Sonestedt: Lund University, Lund, Sweden; Yi Song: Peking University, Beijing, China; Thorkild IA Sørensen: University of Copenhagen, Copenhagen, Denmark; Elin P Sørgjerd: Norwegian University of Science and Technology, Trondheim, Norway; Charles Sossa Jérome: Institut Régional de Santé Publique, Ouidah, Benin; Victoria E Soto-Rojas: Universidad

Icesi, Cali, Colombia; Aïcha Soumaré: University of Bordeaux, Bordeaux, France; Slavica Sovic: University of Zagreb School of Medicine, Zagreb, Croatia; Bente Sparboe-Nilsen: Oslo Metropolitan University, Oslo, Norway; Karen Sparrenberger: Universidade Federal do Rio Grande do Sul, Porto Alegre, Brazil; Angela Spinelli: Istituto Superiore di Sanità, Rome, Italy; Igor Spiroski: Institute of Public Health, Skopje, North Macedonia; Ss. Cyril and Methodius University, Skopje, North Macedonia; Jan A Staessen: University of Leuven, Leuven, Belgium; Hanspeter Stamm: Lamprecht und Stamm Sozialforschung und Beratung AG, Zurich, Switzerland; Maria G Stathopoulou: Institut National de la Santé et de la Recherche Médicale, Nancy, France; Kaspar Staub: University of Zurich, Zurich, Switzerland; Bill Stavreski: Heart Foundation, Melbourne, Australia; Jostein Steene-Johannessen: Norwegian School of Sport Sciences, Oslo, Norway; Peter Stehle: Bonn University, Bonn, Germany; Aryeh D Stein: Emory University, Atlanta, United States; George S Stergiou: Sotiria Hospital, Sotiria, Greece; Jochanan Stessman: Hadassah University Medical Center, Jerusalem, Israel; Ranko Stevanovic: Croatian Institute of Public Health, Zagreb, Croatia; Jutta Stieber: Helmholtz Zentrum München, Munich, Germany; Doris Stöckl: Helmholtz Zentrum München, Munich, Germany; Tanja Stocks: Lund University, Lund, Sweden; Jakub Stokwiszewski: National Institute of Public Health - National Institute of Hygiene, Warsaw, Poland; Ekaterina Stoyanova: Kalina Malina Kindergarten, Pazardjik, Bulgaria; Gareth Stratton: Swansea University, Swansea, United Kingdom; Karien Stronks: University of Amsterdam, Amsterdam, Netherlands; Maria Wany Strufaldi: Federal University of São Paulo, São Paulo, Brazil; Lela Sturua: National Center for Disease Control and Public Health, Tbilisi, Georgia; Ramón Suárez-Medina: National Institute of Hygiene, Epidemiology and Microbiology, Havana, Cuba; Machi Suka: The Jikei University School of Medicine, Tokyo, Japan; Chien-An Sun: Fu Jen Catholic University, Taipei, Taiwan; Johan Sundström: Uppsala University, Uppsala, Sweden; Yn-Tz Sung: The Chinese University of Hong Kong, Hong Kong, China; Jordi Sunyer: Barcelona Institute for Global Health CIBERESP, Barcelona, Spain; Paibul Suriyawongpaisal: Mahidol University, Nakhon Pathom, Thailand; Boyd A Swinburn: University of Auckland, Auckland, New Zealand; Rody G Sy: University of the Philippines, Manila, Philippines; Holly E Syddall: University of Southampton, Southampton, United Kingdom; René Charles Sylva: National Statistical Office, Praia, Cabo Verde; Moyses Szklo: Johns Hopkins Bloomberg School of Public Health, Baltimore, United States; Lucjan Szponar: National Institute of Public Health – National Institute of Hygiene, Warsaw, Poland; E Shyong Tai: National University of Singapore, Singapore, Singapore; Mari-Liis Tammesoo: University of Tartu, Tartu, Estonia; Abdonas Tamosiunas: Lithuanian University of Health Sciences, Kaunas, Lithuania; Eng Joo Tan: Deakin University, Sydney, Australia; Xun Tang: Peking University, Beijing, China; Maya Tanrygulyyeva: Scientific Research Institute of Maternal and Child Health, Ashgabat, Turkmenistan; Frank Tanser: University of Lincoln, Lincoln, United Kingdom; Yong Tao: Peking University, Beijing, China; Mohammed Rasoul Tarawneh: Ministry of Health, Amman, Jordan; Jakob Tarp: Norwegian School of Sport Sciences, Oslo, Norway; Carolina B Tarqui-Mamani: National Institute of Health, Lima, Peru; Radka Taxová Braunerová: Institute of Endocrinology, Prague, Czech Republic; Anne Taylor: University of Adelaide, Adelaide, Australia; Julie Taylor: University College London, London, United Kingdom; Félicité Tchibindat: UNICEF, Niamey, Niger; William R Tebar: Universidade Estadual Paulista, Presidente Prudente, Brazil; Grethe S Tell: University of Bergen, Bergen, Norway; Tania Tello: Universidad Peruana Cayetano Heredia, Lima, Peru; Yih Chung Tham: Singapore Eye Research Institute, Singapore, Singapore; KR Thankappan: Central University of Kerala, Kasaragod, India; Holger Theobald: Karolinska Institutet, Huddinge, Sweden; Xenophon Theodoridis: Aristotle University of Thessaloniki, Thessaloniki, Greece; Lutgarde Thijs: University of Leuven, Leuven, Belgium; Nihal Thomas: Christian Medical College, Vellore, India; Betina H Thuesen: Bispebjerg and Frederiksberg Hospital, Copenhagen, Denmark; Lubica Tichá: Comenius University, Bratislava, Slovakia; Erik J Timmermans: Amsterdam Public Health Research Institute, Amsterdam, Netherlands; Anne Tjonneland: Danish Cancer Society Research Center, Copenhagen, Denmark; Hanna K Tolonen: Finnish Institute for Health and Welfare, Helsinki, Finland; Janne S Tolstrup: University of Southern Denmark, Copenhagen, Denmark; Murat Topbas: Karadeniz Technical University, Trabzon, Turkey; Roman Topór-Madry: Jagiellonian University Medical College, Kraków, Poland; Liv Elin Torheim: Oslo Metropolitan University, Oslo, Norway; María José Tormo: Health Service of Murcia, Murcia, Spain; Michael J Tornaritis: Research and Education

Institute of Child Health, Nicosia, Cyprus; Maties Torrent: Institut d'Investigacio Sanitaria Illes Balears, Menorca, Spain; Laura Torres-Collado: CIBER en Epidemiología y Salud Pública, Alicante, Spain; Stefania Toselli: University of Bologna, Bologna, Italy; Giota Touloumi: National and Kapodistrian University of Athens, Athens, Greece; Pierre Traissac: French National Research Institute for Sustainable Development, Montpellier, France; Thi Tuyet-Hanh Tran: Hanoi University of Public Health, Hanoi, Viet Nam; Dimitrios Trichopoulos: Harvard TH Chan School of Public Health, Boston, United States; Antonia Trichopoulou: Hellenic Health Foundation, Athens, Greece; Oanh TH Trinh: University of Medicine and Pharmacy at Ho Chi Minh City, Ho Chi Minh City, Viet Nam; Atul Trivedi: Government Medical College, Bhavnagar, India; Lechaba Tshepo: Sefako Makgatho Health Science University, Ga-Rankuwa, South Africa; Maria Tsigga: International Hellenic University, Thessaloniki, Greece; Shoichiro Tsugane: National Cancer Center, Tokyo, Japan; Azaliia M Tuliakova: Ufa Eye Research Institute, Ufa, Russian Federation; Marshall K Tulloch-Reid: The University of the West Indies, Kingston, Jamaica; Fikru Tullu: Addis Ababa University, Addis Ababa, Ethiopia; Tomi-Pekka Tuomainen: University of Eastern Finland, Kuopio, Finland; Jaakko Tuomilehto: Finnish Institute for Health and Welfare, Helsinki, Finland; Maria L Turley: Ministry of Health, Wellington, New Zealand; Gilad Twig: Tel-Aviv University, Tel-Aviv, Israel; Hebrew University of Jerusalem, Jerusalem, Israel; Per Tynelius: Karolinska Institutet, Stockholm, Sweden; Themistoklis Tzotzas: Hellenic Medical Association for Obesity, Athens, Greece; Christophe Tzourio: University of Bordeaux, Bordeaux, France; Peter Ueda: Karolinska Institutet, Stockholm, Sweden; Eunice Ugel: Universidad Centro-Occidental Lisandro Alvarado, Barquisimeto, Venezuela; Flora AM Ukoli: Meharry Medical College, Nashville, United States; Hanno Ulmer: Medical University of Innsbruck, Innsbruck, Austria; Belgin Unal: Dokuz Eylul University, Izmir, Turkey; Zhamyila Usupova: Republican Center for Health Promotion, Bishkek, Kyrgyzstan; Hannu MT Uusitalo: University of Tampere Tays Eye Center, Tampere, Finland; Nalan Uysal: Sabiha Gokcen Ilkokulu, Ankara, Turkey; Justina Vaitkeviciute: Lithuanian University of Health Sciences, Kaunas, Lithuania; Gonzalo Valdivia: Pontificia Universidad Católica de Chile, Santiago, Chile; Susana Vale: Polytechnic Institute of Porto, Porto, Portugal; Damaskini Valvi: Icahn School of Medicine at Mount Sinai, New York City, United States; Rob M van Dam: National University of Singapore, Singapore, Singapore; Johan Van der Heyden: Sciensano, Brussels, Belgium; Yvonne T van der Schouw: Utrecht University, Utrecht, Netherlands; Koen Van Herck: Ghent University, Ghent, Belgium; Hoang Van Minh: Hanoi University of Public Health, Hanoi, Viet Nam; Natasja M Van Schoor: VU University Medical Center, Amsterdam, Netherlands; Irene GM van Valkengoed: University of Amsterdam, Amsterdam, Netherlands; Dirk Vanderschueren: Katholieke Universiteit Leuven, Leuven, Belgium; Diego Vanuzzo: Cardiovascular Prevention Centre Udine, Udine, Italy; Anette Varbo: Copenhagen University Hospital, Copenhagen, Denmark; University of Copenhagen, Copenhagen, Denmark; Gregorio Varela-Moreiras: Universidad CEU San Pablo, Madrid, Spain; Patricia Varona-Pérez: National Institute of Hygiene, Epidemiology and Microbiology, Havana, Cuba; Senthil K Vasan: University of Southampton, Southampton, United Kingdom; Tomas Vega: Consejería de Sanidad Junta de Castilla y León, Valladolid, Spain; Toomas Veidebaum: National Institute for Health Development, Tallinn, Estonia; Gustavo Velasquez-Melendez: Universidade Federal de Minas Gerais, Belo Horizonte, Brazil; Biruta Velika: Centre for Disease Prevention and Control, Riga, Latvia; Giovanni Veronesi: University of Insubria, Varese, Italy; WM Monique Verschuren: National Institute for Public Health and the Environment, Bilthoven, Netherlands; Cesar G Victora: Federal University of Pelotas, Pelotas, Brazil; Giovanni Viegi: National Research Council, Pisa, Italy; Lucie Viet: National Institute for Public Health and the Environment, Bilthoven, Netherlands; Salvador Villalpando: National Institute of Public Health, Cuernavaca, Mexico; Paolo Vineis: Imperial College London, London, United Kingdom; Jesus Vioque: University Miguel Hernandez, Alicante, Spain; Jyrki K Virtanen: University of Eastern Finland, Kuopio, Finland; Marjolein Visser: Vrije Universiteit Amsterdam, Amsterdam, Netherlands; Sophie Visvikis-Siest: Institut National de la Santé et de la Recherche Médicale, Nancy, France; Bharathi Viswanathan: Ministry of Health, Victoria, Seychelles; Mihaela Vladulescu: Sunflower Nursery School, Craiova, Romania; Tiina Vlasoff: North Karelia Center for Public Health, Joensuu, Finland; Dorja Vocanec: University of Zagreb School of Medicine, Zagreb, Croatia; Peter Vollenweider: Lausanne University Hospital, Lausanne, Switzerland; Henry Völzke: University Medicine Greifswald, Greifswald, Germany; Ari Voutilainen: University of Eastern Finland,

Kuopio, Finland; Sari Voutilainen: University of Eastern Finland, Kuopio, Finland; Martine Vrijheid: Barcelona Institute for Global Health CIBERESP, Barcelona, Spain; Tanja GM Vrijkotte: University Medical Centers, Groningen, Netherlands; University of Amsterdam, Amsterdam, Netherlands; Alisha N Wade: University of the Witwatersrand, Johannesburg, South Africa; Aline Wagner: University of Strasbourg, Strasbourg, France; Thomas Waldhör: Medical University of Vienna, Vienna, Austria; Janette Walton: Cork Institute of Technology, Cork, Ireland; Elvis OA Wambiya: African Population and Health Research Center, Nairobi, Kenya; Wan Mohamad Wan Bebakar: Universiti Sains Malaysia, Kelantan, Malaysia; Wan Nazaimoon Wan Mohamud: Institute for Medical Research, Kuala Lumpur, Malaysia; Rildo de Souza Wanderley Júnior: University of Pernambuco, Recife, Brazil; Ming-Dong Wang: Public Health Agency of Canada, Ottawa, Canada; Ningli Wang: Capital Medical University Beijing Tongren Hospital, Beijing, China; Qian Wang: Xinjiang Medical University, Urumqi, China; Xiangjun Wang: Shanghai Educational Development Co. Ltd., Shanghai, China; Ya Xing Wang: Capital Medical University, Beijing, China; Ying-Wei Wang: Ministry of Health and Welfare, Taipei, Taiwan; S Goya Wannamethee: University College London, London, United Kingdom; Nicholas Wareham: University of Cambridge, Cambridge, United Kingdom; Adelheid Weber: Federal Ministry of Social Affairs, Health, Care and Consumer Protection, Vienna, Austria; Niels Wedderkopp: University of Southern Denmark, Odense, Denmark; Deepa Weerasekera: Ministry of Health, Auckland, New Zealand; Daniel Weghuber: Paracelsus Medical University, Salzburg, Austria; Wenbin Wei: Capital Medical University, Beijing, China; Aneta Weres: University of Rzeszów, Rzeszów, Poland; Bo Werner: Örebro University, Örebro, Sweden; Peter H Whincup: St George's, University of London, London, United Kingdom; Kurt Widhalm: Medical University of Vienna, Vienna, Austria; Indah S Widyahening: Universitas Indonesia, Jakarta, Indonesia; Andrzej Wiecek: Medical University of Silesia, Katowice, Poland; Rainford J Wilks: The University of the West Indies, Kingston, Jamaica; Johann Willeit: Medical University of Innsbruck, Innsbruck, Austria; Peter Willeit: Medical University of Innsbruck, Innsbruck, Austria; Julianne Williams: World Health Organization Regional Office for Europe, Moscow, Russian Federation; Tom Wilsgaard: UiT The Arctic University of Norway, Tromsø, Norway; Bogdan Wojtyniak: National Institute of Public Health - National Institute of Hygiene, Warsaw, Poland; Roy A Wong-McClure: Caja Costarricense de Seguro Social, San José, Costa Rica; Andrew Wong: University College London, London, United Kingdom; Jyh Eiin Wong: Universiti Kebangsaan Malaysia, Kuala Lumpur, Malaysia; Tien Yin Wong: Duke-NUS Medical School, Singapore, Singapore; Jean Woo: The Chinese University of Hong Kong, Hong Kong, China; Mark Woodward: University of New South Wales, Sydney, Australia; University of Oxford, Oxford, United Kingdom; Frederick C Wu: University of Manchester, Manchester, United Kingdom; Jianfeng Wu: Shandong University of Traditional Chinese Medicine, Jinan, China; Li Juan Wu: Capital Medical University, Beijing, China; Shouling Wu: Kailuan General Hospital, Tangshan, China; Haiquan Xu: Institute of Food and Nutrition Development of Ministry of Agriculture and Rural Affairs, Beijing, China; Liang Xu: Beijing Institute of Ophthalmology, Beijing, China; Nor Azwany Yaacob: Universiti Sains Malaysia, Kelantan, Malaysia; Uruwan Yamborisut: Mahidol University, Nakhon Pathom, Thailand; Weili Yan: Children's Hospital of Fudan University, Shanghai, China; Ling Yang: University of Oxford, Oxford, United Kingdom; Xiaoguang Yang: Chinese Center for Disease Control and Prevention, Beijing, China; Yang Yang: Shanghai Educational Development Co. Ltd, Shanghai, China; Nazan Yardim: Ministry of Health, Ankara, Turkey; Mehdi Yaseri: Shahid Beheshti University of Medical Sciences, Tehran, Islamic Republic of Iran; Tabara Yasuharu: Kyoto University, Kyoto, Japan; Xingwang Ye: University of Chinese Academy of Sciences, Shanghai, China; Panayiotis K Yiallouros: University of Cyprus, Nicosia, Cyprus; Moein Yoosefi: Non-Communicable Diseases Research Center, Tehran, Islamic Republic of Iran; Akihiro Yoshihara: Niigata University, Niigata, Japan; Qi Sheng You: Capital Medical University, Beijing, China; San-Lin You: Fu Jen Catholic University, Taipei, Taiwan; Novie O Younger-Coleman: The University of the West Indies, Kingston, Jamaica; Safiah Md Yusof: International Medical University, Shah Alam, Malaysia; Ahmad Faudzi Yusoff: Ministry of Health, Kuala Lumpur, Malaysia; Luciana Zaccagni: University of Ferrara, Ferrara, Italy; Vassilis Zafiropulos: Hellenic Mediterranean University, Heraklion, Greece; Ahmad A Zainuddin: Ministry of Health, Kuala Lumpur, Malaysia; Seyed Rasoul Zakavi: Mashhad University of Medical Sciences, Mashhad, Islamic Republic of Iran; Farhad Zamani: Iran University of Medical Sciences, Tehran, Islamic

Republic of Iran; Sabina Zambon: University of Padua, Padua, Italy; Antonis Zampelas: Agricultural University of Athens, Athens, Greece; Hana Zamrazilová: Institute of Endocrinology, Prague, Czech Republic; Maria Elisa Zapata: Centro de Estudios sobre Nutrición Infantil, Buenos Aires, Argentina; Abdul Hamid Zargar: Center for Diabetes and Endocrine Care, Srinagar, India; Ko Ko Zaw: University of Public Health, Yangon, Myanmar; Tomasz Zdrojewski: Medical University of Gdansk, Gdansk, Poland; Kristyna Zejglicova: National Institute of Public Health, Prague, Czech Republic; Tajana Zeljkovic Vrkic: University Hospital Center Zagreb, Zagreb, Croatia; Yi Zeng: Peking University, Beijing, China; Duke University, Durham, United States; Luxia Zhang: Peking University First Hospital, Beijing, China; Zhen-Yu Zhang: University of Leuven, Leuven, Belgium; Dong Zhao: Capital Medical University Beijing An Zhen Hospital, Beijing, China; Ming-Hui Zhao: Peking University First Hospital, Beijing, China; Wenhua Zhao: Chinese Center for Disease Control and Prevention, Beijing, China; Shiqi Zhen: Jiangsu Provincial Center for Disease Control and Prevention, Nanjing, China; Wei Zheng: Vanderbilt University, Nashville, United States; Yingfeng Zheng: Sun Yat-sen University, Guangzhou, China; Bekbolat Zholdin: West Kazakhstan Medical University, Aktobe, Kazakhstan; Maigeng Zhou: Chinese Center for Disease Control and Prevention, Beijing, China; Dan Zhu: Inner Mongolia Medical University, Hohhot, China; Marie Zins: Institut National de la Santé et de la Recherche Médicale, Villejuif, France; Paris University, Paris, France; Emanuel Zitt: Agency for Preventive and Social Medicine, Bregenz, Austria; Yanina Zocalo: Universidad de la República, Montevideo, Uruguay; Julio Zuñiga Cisneros: Instituto Conmemorativo Gorgas de Estudios de la Salud, Panama City, Panama; Monika Zuziak: Przedszkole No. 81, Warsaw, Poland; Majid Ezzati: Imperial College London, London, United Kingdom; University of Ghana, Accra, Ghana; Sarah Filippi: Imperial College London, London, United Kingdom

## Funding

| Funder | Author |
|---|---|
| Wellcome Trust | Majid Ezzati |
| Medical Research Council | Maria LC Iurilli |

The funders had no role in study design, data collection and interpretation, or the decision to submit the work for publication.

## Author contributions

NCD-RisC, Conceptualization, Resources, Data curation, Software, Formal analysis, Supervision, Funding acquisition, Validation, Investigation, Visualization, Methodology, Writing - original draft, Writing - review and editing, Project administration

## Author ORCIDs

Maria LC Iurilli  https://orcid.org/0000-0003-0409-1635
Bin Zhou  https://orcid.org/0000-0002-1741-8628
Majid Ezzati  https://orcid.org/0000-0002-2109-8081

## Decision letter and Author response

Decision letter https://doi.org/10.7554/eLife.60060.sa1
Author response https://doi.org/10.7554/eLife.60060.sa2

# Additional files

## Supplementary files

• Supplementary file 1. Coefficients of the regression of probit-transformed prevalence of underweight, obesity, and severe obesity in women on mean body mass index.

• Supplementary file 2. Coefficients of the regression of probit-transformed prevalence of underweight, obesity, and severe obesity in men on mean body mass index.

• Supplementary file 3. List of analysis regions and countries in each region.

- Supplementary file 4. List of data sources used in the analysis and their characteristics.

- Supplementary file 5. Mean body mass index estimates (kg/m$^2$) in 1985 and 2016 stratified by region, gender, and age group.

- Source data 1. Data from studies in the Non-Communicable Disease Risk Factor Collaboration database conducted from 1985 to 2019 with participants aged 20–79 years old.

- Source code 1. Liner regression model used for the analysis of the association of prevalence of underweight, total obesity, and severe obesity with mean body mass index.

- Transparent reporting form

### Data availability

Names and characteristics of data sources included in this pooling analysis are listed in Supplementary file 4. Of these data, some are from public sources, for which we have provided the data in Source data 1. Others are from individual researchers and/or from government and international agencies; these should be requested from the data holders on a study to study basis using the information in Source data 1.

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
