## [Decision Letter]

**Acceptance summary:**

This paper will be of interest to public health scientists and practitioners concerned with non-communicable diseases related to body weight. The authors have compiled global data on body mass index (BMI) and body weight classification to assess the association between changes in mean BMI and weight classification over three decades. These data will also be of value for policy reviews and evidence syntheses.

**Decision letter after peer review:**

Thank you for submitting your article "Heterogeneous contributions of change in population mean body-mass index to changes in obesity and underweight" for consideration by *eLife*. Your article has been reviewed by three peer reviewers, and the evaluation has been overseen by a Reviewing Editor and a Senior Editor. The following individual involved in review of your submission has agreed to reveal their identity: Simon Capewell (Reviewer #1).

As is customary in *eLife*, the reviewers have discussed their critiques with one another. What follows below is the Reviewing Editor's edited compilation of the essential and ancillary points provided by reviewers in their critiques and in their interaction post-review. Please submit a revised version that addresses these concerns directly. Although we expect that you will address these comments in your response letter we also need to see the corresponding revision in the text of the manuscript. Some of the reviewers' comments may seem to be simple queries or challenges that do not prompt revisions to the text. Please keep in mind, however, that readers may have the same perspective as the reviewers. Therefore, it is essential that you attempt to amend or expand the text to clarify the narrative accordingly.

Because many researchers have temporarily lost access to the labs, we will give authors as much time as they need to submit revised manuscripts. We are also offering, if you choose, to post the manuscript to bioRxiv (if it is not already there) along with this decision letter and a formal designation that the manuscript is "in revision at *eLife*". Please let us know if you would like to pursue this option. (If your work is more suitable for medRxiv, you will need to post the preprint yourself, as the mechanisms for us to do so are still in development.)

Summary:

This potentially important analysis of three-decade trends in BMI distribution involved an impressive international collaboration with a dataset comprising 2,675 population-based studies with over 160 million participants. The prevalence of underweight decreased while obesity increased in most regions, being largely driven by shifts in the distribution of BMI, with only small contributions from changes in the shape of the distribution. Despite overall falls, substantial levels of underweight persisted in sub-Saharan Africa and parts of Asia. Overall, the paper makes a good descriptive contribution.

Essential revisions:

The authors have collated measured data from 9 global regions to examine changes in mean BMI and prevalence of underweight, overweight, and severe overweight categories among adults between 1985 and 2016. The key questions are how much change in mean BMI explains changing category prevalences over the period, and how these effects vary by region. The analysis is potentially informative and interesting, but concerns were raised that the best approach to answering the question is not used, primarily due to reliance upon means and categorical outcomes to assess distribution shape changes. It is not apparent how the data and interpretation would substantially change understanding or practice.

One main comment is that the conclusion (i.e., the heterogeneity of the association between mean and distribution) is not quite clearly communicated, and secondly what does this mean? The authors should expand substantially on detailing the nature of this heterogeneity and its implications at the country-specific level perhaps by reviewing the epidemiologic evolution of the BMI. Without that the paper does not add much to the "overall" knowledge on this subject which has been explored before as a critical observation that was missed. While the authors cite those papers, besides taking that to scale with more data there is not much that is novel in terms of the idea per se. Another issue is the "heterogeneity" in the data source itself that introduces considerable measurement noise which, in turn, can influence the variance or change in the parameter with respect to the distribution.

1) Given access to the individual-level data, why were distribution characteristics not examined in the continuous data, e.g. using percentiles? Using classification data adds another source of error (misclassification) that depends on choice of cut-points (recognizing that consistent criteria were used across studies). If the authors did not have access to individual ht/wt data, but only mean BMI and prevalences of weight categories from each of the data sources, this should be clarified.

2) Discussion – related to the above, is it unclear why visual examination of distribution changes was not possible. With 2 sexes, 2 age groups, and 9 regions, the numbers do not seem excessive to examine distributions.

3) The authors discuss how changes in prevalence are accounted for by changes in mean BMI, but because the mean is disproportionately affected by extreme values, could causality not go in the other direction with the mean being affected by changes in the tails?

4) Discussion, first sentence of the last paragraph – this sentence seems to state a tautology. The remainder of the paragraph, while perhaps noteworthy and true, does not seem to be data-based from this analysis. Perhaps a clearer link to the data is needed.

5) Subsection “Statistical methods”, last paragraph – again, related to issues of what data were available, this paragraph states mean BMI were derived from a previous publication and not calculated from the pooled data.

---

## [Author Response]

Essential revisions:The authors have collated measured data from 9 global regions to examine changes in mean BMI and prevalence of underweight, overweight, and severe overweight categories among adults between 1985 and 2016. The key questions are how much change in mean BMI explains changing category prevalences over the period, and how these effects vary by region. The analysis is potentially informative and interesting, but concerns were raised that the best approach to answering the question is not used, primarily due to reliance upon means and categorical outcomes to assess distribution shape changes. It is not apparent how the data and interpretation would substantially change understanding or practice.

As the reviewers and Editors correctly point out, it is possible to describe distributional change through other approaches – for example reporting changes in its second and third moments (standard deviation and skewness) or in different quantiles of distribution. However, these approaches generate results that are internal to the distribution for each study. As a result, the results are neither comparable across regions, nor map to clinical and epidemiological measures such as underweight and obesity. As we state in the paper’s Introduction, our aim was to understand how much change in underweight and obesity, which are commonly used to make clinical and public health decisions, are driven by change in population mean vs. change in the shape of the distribution. As we state in the Introduction, understanding the relative contributions of these mechanisms has public health implications. Specifically, our finding that change in mean BMI contributed to the majority of change in the prevalence of underweight, obesity and severe obesity argues for population-wide approaches as opposed to clinical management of high-risk individuals (subsection “Study design”).

One main comment is that the conclusion (i.e, the heterogeneity of the association between mean and distribution) is not quite clearly communicated, and secondly what does this mean? The authors should expand substantially on detailing the nature of this heterogeneity and its implications at the country-specific level perhaps by reviewing the epidemiologic evolution of the BMI. Without that the paper does not add much to the "overall" knowledge on this subject which has been explored before as a critical observation that was missed. While the authors cite those papers, besides taking that to scale with more data there is not much that is novel in terms of the idea per se. Another issue is the "heterogeneity" in the data source itself that introduces considerable measurement noise which, in turn, can influence the variance or change in the parameter with respect to the distribution.

As the comment correctly states, previous studies have used a small number of surveys in one or a few countries and reported, graphically or with summary statistics, how the distribution may have shifted or changed. Using a small number of data sources gives an impression of precision in measured change compared to using a larger number of data sources. This is not the same as less “measurement noise” but rather overlooking the fact that each single study is only a sample of the world. By pooling a large number of data sources, we get much closer to more valid estimates of how much distributional shift and change have contributed to changing prevalence.

Our paper is novel and important because of both the unprecedented scale of data, and because it is (to our knowledge) the first ever to report and compare the contributions of distributional shift vs. change to changes in underweight and obesity prevalence for all regions of the world. The specific size of these contributions, and their variations across regions, is a major new result.

In addition to these contributions, as raised by this comment, our analysis allowed for the relationship between underweight/obesity prevalence and mean BMI to vary across regions by including interactions between region and mean BMI (subsection “Statistical methods”). These results are in Supplementary files 1 and 2 of the revised paper but had not been explicitly stated in the paper. We have added a new section on the association between prevalence and mean, and its inter-region variability (subsection “Associations of underweight, obesity and severe obesity prevalence with mean BMI”).

1) Given access to the individual-level data, why were distribution characteristics not examined in the continuous data, e.g. using percentiles? Using classification data adds another source of error (misclassification) that depends on choice of cut-points (recognizing that consistent criteria were used across studies). If the authors did not have access to individual ht/wt data, but only mean BMI and prevalences of weight categories from each of the data sources, this should be clarified.

As stated above, we could have used continuous data to report changes in its second and third moments or in different quantiles of distribution. However, these approaches generate results that are internal to the distribution for each region. While these alternatives are statistically as valid as our approach, they are neither comparable across regions, nor do they map to clinical and epidemiological measures such as underweight and obesity. In contrast, underweight and obesity are measures that are used for clinical and public health purposes, and reported in virtually every policy report. If of interest, we would be happy to also report some information on changes in measures such as standard deviation (Author response image 1) and skewness (Author response image 2) emphasising with the above caveat that these are not directly related to underweight and obesity.

**Author response image 1. respfig1:** Standard deviation of BMI from 1985 to 2019 by sex and age group. Each point shows one age-sex group in one study. The size of each point is proportional to its sample size. Colour for each point indicates the region it is from.

**Author response image 2. respfig2:** Skewness of BMI from 1985 to 2019 by sex and age group. Each point shows one age-sex group in one study. The size of each point is proportional to its sample size. Colour for each point indicates the region it is from.

2) Discussion – related to the above, is it unclear why visual examination of distribution changes was not possible. With 2 sexes, 2 age groups, and 9 regions, the numbers do not seem excessive to examine distributions.

Studies that visually inspect distributions often use 2 or 3 data sources. However, our usage of over 2,800 makes it impossible to make visual comparisons as seen in Author response image 3 for one age group.

**Author response image 3. respfig3:** Distributions of BMI in NCD-RisC studies for 50-59-year old by sex and region. Each curve is a single study and the colour indicates its year from 1985 to 2019.

3) The authors discuss how changes in prevalence are accounted for by changes in mean BMI, but because the mean is disproportionately affected by extreme values, could causality not go in the other direction with the mean being affected by changes in the tails?

Author response image 4 shows a comparison of mean and median in those studies for which we had data on median (~70% of all the studies used for the analysis). As seen in the figure, they are highly correlated (correlation coefficient ≥ 0.98 for all age groups and genders) indicating that while this issue is theoretically possible, in practice it does not affect our analysis.

**Author response image 4. respfig4:** Comparison of mean and median BMI by sex and age group. Each point represents a sex-age group in a study conducted between 1985 and 2019.

4) Discussion, first sentence of the last paragraph – this sentence seems to state a tautology. The remainder of the paragraph, while perhaps noteworthy and true, does not seem to be data-based from this analysis. Perhaps a clearer link to the data is needed.

We have reworded to more directly relate to our results as appropriately suggested (Discussion).

5) Subsection “Statistical methods”, last paragraph – again, related to issues of what data were available, this paragraph states mean BMI were derived from a previous publication and not calculated from the pooled data.

This is a good point which we have clarified at the beginning of the subsection ”Data sources”. In brief, we used original data – directly in the first step of the analysis and after fitting a Bayesian model in the second step.